# Hierarchical temporal prediction captures motion processing along the visual pathway

**Yosef Singer\*†, Luke Taylor†, Ben DB Willmore, Andrew J King\*, Nicol S Harper\***

Department of Physiology, Anatomy and Genetics, University of Oxford, Oxford, United Kingdom

**Abstract** Visual neurons respond selectively to features that become increasingly complex from the eyes to the cortex. Retinal neurons prefer flashing spots of light, primary visual cortical (V1) neurons prefer moving bars, and those in higher cortical areas favor complex features like moving textures. Previously, we showed that V1 simple cell tuning can be accounted for by a basic model implementing temporal prediction – representing features that predict future sensory input from past input (Singer et al., 2018). Here, we show that hierarchical application of temporal prediction can capture how tuning properties change across at least two levels of the visual system. This suggests that the brain does not efficiently represent all incoming information; instead, it selectively represents sensory inputs that help in predicting the future. When applied hierarchically, temporal prediction extracts time-varying features that depend on increasingly high-level statistics of the sensory input.

**\*For correspondence:**
yossing@gmail.com (YS);
andrew.king@dpag.ox.ac.uk (AJK);
nicol.harper@dpag.ox.ac.uk (NSH)

†These authors contributed equally to this work

## Editor's evaluation

This valuable work shows similarities between a multilayer, convolutional neural network trained to predict its next input and physiological features of visual processing in the brain. These solid results build on the authors' previous work and compare the match to real visual processing obtained by a hierarchical predictive network to that obtained by several other popular artificial neural networks. This work will be of interest to systems neuroscientists as well as computer scientists looking to make connections between normative theories of neural organization and training objectives in machine learning.

## Introduction

The temporal prediction (*Singer et al., 2018*) framework posits that sensory systems are optimized to represent features in natural stimuli that enable prediction of future sensory input. This would be useful for guiding future action, uncovering underlying variables, and discarding irrelevant information (*Bialek et al., 2001*; *Singer et al., 2018*). Temporal prediction relates to a class of principles, such as the predictive information bottleneck (*Bialek et al., 2001*; *Chalk et al., 2018*; *Salisbury and Palmer, 2016*) and slow feature analysis (*Berkes and Wiskott, 2005*), that similarly involve selectively encoding *only* features that are efficiently predictive of the future. This class of principles has been contrasted (*Chalk et al., 2018*; *Salisbury and Palmer, 2016*) with other principles that are more typically used to explain sensory coding – efficient coding (*Barlow, 1961*; *Bell and Sejnowski, 1997*), sparse coding (*Olshausen and Field, 1996*), and predictive coding (*Huang and Rao, 2011*; *Rao and Ballard, 1999*) – that aim instead to efficiently represent *all* current and perhaps past input. Although these principles have been successful in accounting for various visual receptive field (RF) properties

**Figure 1.** An illustration of the general architecture of the hierarchical temporal prediction model. (**a**) Architecture of the hierarchical temporal prediction model. Each stack of the model (labeled 'predictive feature extractor') receives input from the stack below and finds features of its input that are predictive of its future input, with the lowest stack predicting the raw natural inputs. The predictive signal is fed forward to the next layer. (**b**) Operations performed by each stack of the temporal prediction model. Each stack receives a time-varying input vector $\mathbf{u}(t)$, which in all but the first stack is the hidden unit activity vector $\mathbf{h}(t)$ from the lower stack. For the first stack, the input $\mathbf{u}(t)$ is raw visual input (i.e. video). The input, also with some additional delayed input, passes through feedforward input weight matrix $\mathbf{W}$ and undergoes a nonlinear transformation $\mathbf{f}()$ to generate hidden unit activity $\mathbf{h}(t)$. Each stack is optimized so that a linear transformation $\mathbf{M}$ from its hidden unit activity $\mathbf{h}(t)$, and then a delay, generates an accurate estimate $\hat{\mathbf{u}}_t(t-1)$ of its future input (at time $t$). Because of the delay, the predicted input is in the future relative to the input that generated the hidden unit activity. This delay is shown at the end of processing, but is likely distributed throughout the system. For each stack, the prediction error is only used to train the input and output weights $\mathbf{W}$ and $\mathbf{M}$. The prediction error can equivalently be written as $\mathbf{e}(t+1) = \mathbf{u}(t+1) - \hat{\mathbf{u}}_{t+1}(t)$. To see how this hierarchical temporal prediction model differs from predictive coding (*Rao and Ballard, 1999*) and PredNet (*Lotter et al., 2020*) see *Figure 1—figure supplement 1*.

The online version of this article includes the following figure supplement(s) for figure 1:

**Figure supplement 1.** Architecture and operations performed by temporal prediction, predictive coding and PredNet models.

in V1 (*Bell and Sejnowski, 1997*; *Berkes and Wiskott, 2005*; *Chalk et al., 2018*; *Olshausen and Field, 1996*; *Rao and Ballard, 1999*; *Singer et al., 2018*), no single principle has so far been able to explain the diverse spatiotemporal tuning involved in motion processing that emerges along the visual stream, from retina to V1 to the extrastriate middle temporal area (MT).

Any general principle of visual encoding needs to explain temporal aspects of neural tuning – the encoding of visual scenes in motion rather than static images. It is also important that any general principle is largely unsupervised. Some features of the visual system have been reproduced by deep supervised network models optimized for image classification using large labeled datasets (e.g. images labeled as cat, dog, car; *Yamins and DiCarlo, 2016*). While these models can help to explain the RF properties of the likely hard-wired retina (*Lindsey et al., 2019*), they are less informative if neuronal tuning is influenced by experience, as in cortex, since most sensory input is unlabeled except for sporadic reinforcement signals. The temporal prediction approach is unsupervised (i.e. it requires no labeled data), and inherently applies to the temporal domain.

Several theoretical studies have proposed prediction of future input as a primary function of neural systems (*Bialek et al., 2001*; *Hawkins and Blakeslee, 2004*; *O'Reilly et al., 2014*; *Softky, 1996*). However, models incorporating temporal prediction that explicitly attempt, with varying degrees of success, to explain features of sensory representations from natural inputs have only very recently started to emerge (*Chalk et al., 2018*; *Lotter et al., 2020*; *Ocko et al., 2018*; *Palm, 2012*; *Singer et al., 2018*). We have previously shown that a simple non-hierarchical model instantiating temporal prediction can account for temporal aspects of V1 simple cell RFs (*Singer et al., 2018*). This model finds features in its inputs that are predictive of future input. However, the pattern of occurrence of these particular predictive features over time may itself have a predictable structure. Hence a second temporal prediction model could be stacked atop the first to find this structure. This could be done repeatedly so as to find the predictable structure of increasingly high-level statistics of the input (*Figure 1a*).

Here, we have developed such a hierarchical form of the temporal prediction model that predicts increasingly high-level statistics of natural dynamic visual input (*Figure 1*). This hierarchical model stands in contrast to the influential hierarchical predictive-coding related approaches that transmit forward the prediction *error* (*Lotter et al., 2020*; *Rao and Ballard, 1999*), rather than passing forward

the predictive *signal* (*Figure 1—figure supplement 1*). Our model accounts not only for linear tuning properties of V1 simple cells, as in previous non-hierarchical temporal prediction models (*Chalk et al., 2018*; *Singer et al., 2018*), but also for the diversity of linear and non-linear spatiotemporal tuning properties that emerge along the visual pathway. In particular, the temporal tuning properties in successive hierarchical stages of the model progress from those resembling magnocellular and parvocellular neurons at early levels of visual processing, to direction-selective simple and complex cells in V1, and finally to a few units that are sensitive to two-dimensional features of motion, as seen in pattern-selective cells in higher visual cortex. The capacity of this model to explain spatiotemporal tuning properties at several levels of the visual pathway using iterated application of a single process, suggests that optimization for temporal prediction may be a fundamental principle of sensory neural processing.

## Results
### The hierarchical temporal prediction model

We instantiated temporal prediction as a hierarchical model consisting of stacked feedforward single-hidden-layer convolutional neural networks (*Figure 2*). Image statistics tend to be similar across space, so convolution allows the networks to be efficiently trained and implemented while providing units in deeper stacks the capacity to receive input from a wide span of the image. The first stack was trained using backpropagation to predict the immediate future frame (40ms) of unfiltered natural video inputs from the previous 5 frames (200ms). Each subsequent stack was then trained to predict the future hidden-unit activity of the stack below it from the past activity in response to the natural video inputs. $L_1$ regularization was applied to the weights of each stack, akin to a constraint on neural 'wiring'.

The video inputs consisted of animals moving in natural environments, filmed with panning, still and occasionally zooming cameras. Unlike in *Singer et al., 2018*, we did not bandpass filter the videos in the manner of *Olshausen and Field, 1997*. The four stacks respectively used 50, 100, 200, and 400 convolutional units, each having a particular convolutional kernel. The convolutional units can also be seen as a set of hidden units, with each hidden unit having the same weights relative to its position in the space of the video frame. However, because we only use valid convolutions, the number of spatial positions decreases with each stack (see *Table 1*, Methods). This results in 14,450, 22,500, 33,800, and 48,400 hidden units over all spatial positions in each stack, respectively. Hereafter, the term model units refers to the properties of the convolutional units of the model, unless noted otherwise.

After training, we examined the response properties of the units in the model and compared them to measured response properties in the visual processing pathway. The trained model is very rich, with diverse properties. Here, we have chosen to focus mostly on those properties relating to motion sensitivity. However, we expect there to be many other response properties of the model units that could also be examined and compared to other aspects of the biology. The construction of this model has not been tailored to any specific species and aims to capture general features common across mammals. However, for consistency, throughout the Results, we compare the tuning properties of our model units mostly to those of macaque visual neurons.

### Temporal prediction of natural inputs produces retinal-like units

After training, we first examined the input weights of the units in the first stack. Each hidden unit can be viewed as a linear non-linear (LN) model (*Chichilnisky, 2001*; *Simoncelli et al., 2004*), as commonly used to describe neuronal RFs. With $L_1$ regularization slightly above the optimum for prediction, the RFs of the units showed spatially localized, center-surround tuning with a decaying temporal envelope, characteristic of retinal and lateral geniculate nucleus (LGN) neurons (*Barlow, 1953*; *Kuffler, 1953*; *Shapley and Perry, 1986*). The model units' RFs have either an excitatory (ON) or inhibitory (OFF) blob-like structure at the 0 ms time-step, often with a surround of opposing sign in the same or previous time-step (*Figure 3a*). Both ON and OFF units can have either small RFs that do not change polarity over time (*Figure 3a*, units 1–4; *Figure 3b*, bottom left), or large RFs that switch polarity over time (*Figure 3a*, units 5–6; *Figure 3b*, top right). This is reminiscent of the four main cell types in the primate retina and LGN: the parvocellular pathway ON/OFF neurons and the more change-sensitive magnocellular pathway ON/OFF neurons, respectively (*Shapley and Perry, 1986*).

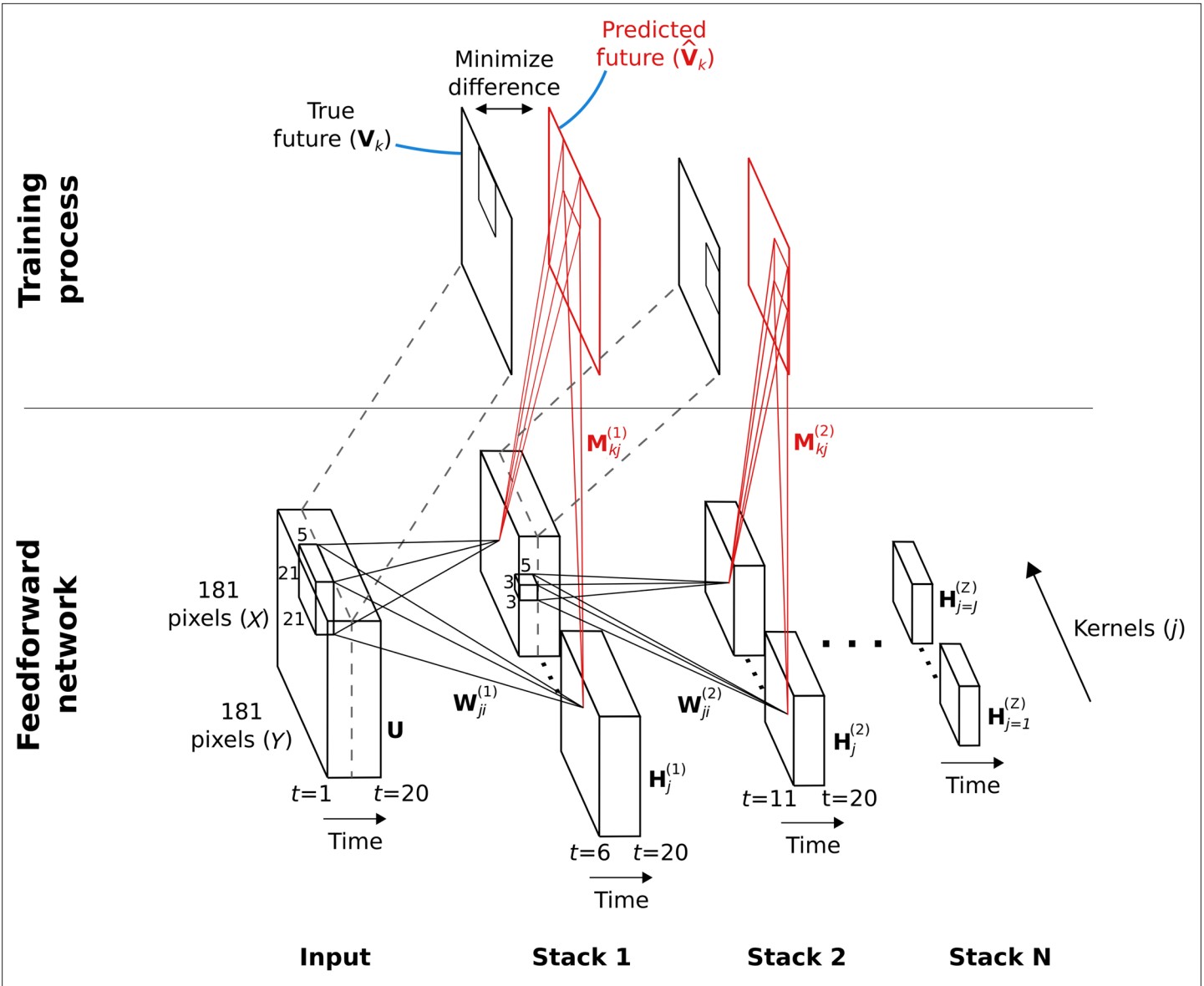

**Figure 2.** Hierarchical temporal prediction model in detail, with convolution. Schematic of model architecture, illustrating the stacking and convolution. This model is essentially the same as the model in *Figure 1* but using convolutional networks; the notation used here is slightly different from *Figure 1* due to this. Each stack is a single hidden-layer feedforward convolutional network, which is trained to predict the future time-step of its input from the previous 5 time-steps. Each stack consists of a bank of $J$ convolutional hidden units, where a convolutional hidden unit is a set of hidden units that each have the same weights (the same convolution kernels). The first stack is trained to predict future pixels of natural video inputs, $\mathbf{U}$, from their past. Subsequent stacks are trained to predict future time-steps of the hidden-layer activity in the stack below, $\mathbf{H}$, based on their past responses to the same natural video inputs. For each convolutional hidden unit $j$ in a stack, its input $\mathbf{U}_i$ from convolutional hidden unit $i$ in the stack below is convolved with an input weight kernel $\mathbf{W}_{ji}$, summed over $i$, and then rectified, to produce its hidden unit activity $\mathbf{H}_j$. This activity is next convolved with output weight kernels $\mathbf{M}_{kj}$ and summed over $j$ to provide predictions $\hat{\mathbf{V}}_k$ of the immediate future input $\mathbf{V}_k$ of each convolutional hidden unit, $k$, in the stack below. Note, $\mathbf{V}_k$ is just $\mathbf{U}_i$ shifted one time step into the future, with $k = i$. The weights ($\mathbf{W}$, $\mathbf{M}$) of each stack are then trained by backpropagation to minimize the difference between the predicted future $\hat{\mathbf{V}}_k$ and the true future, $\mathbf{V}_k$. Each stack is trained separately, starting with the lowest stack, which, once trained, provides input for the next stack, which is then trained, and so on. The input and hidden stages of all the stacks when stacked together form a deep feedforward network. Note, to avoid clutter, only one time slice $\hat{\mathbf{V}}_k$ and $\mathbf{V}_k$ are shown. In the convolution model presented here, every bold capitalized variable is a 3D-tensor over space ($x$), space ($y$) and time ($t$), and every italicized variable is a scalar, either a real number or an integer. The bracketed superscript on variables denotes the stack number $z$ up to top stack $Z$.

**Table 1.** Model parameter settings for each stack.

| Stack | Input size (X,Y,T,I) | Hidden layer size | Kernel size (X',Y',T') | Spatial and temporal extent | Stride (s₁,s₂,s₃) | Number of convolutional hidden units | Learning rate (α) | L₁ regularization strength (λ) |
|---|---|---|---|---|---|---|---|---|
| 1 | 181x18x20x1 | 17x17x16x50 | 21x21x5 | 21x21x5 | 10,10,1 | 50 | $10^{-2}$ | $10^{-4.5}$ |
| 2 | 17x17x16x50 | 15x15x12x100 | 3x3x5 | 41x41x9 | 1,1,1 | 100 | $10^{-4}$ | $10^{-6}$ |
| 3 | 15x15x12x100 | 13x13x8x200 | 3x3x5 | 61x61x13 | 1,1,1 | 200 | $10^{-4}$ | $10^{-6}$ |
| 4 | 13x13x8x200 | 11x11x4x400 | 3x3x5 | 81x81x17 | 1,1,1 | 400 | $10^{-4}$ | $10^{-6}$ |

Interestingly, simply decreasing $L_1$-regularization strength causes the model RFs to change from center-surround tuning to Gabor-like tuning, resembling localized, oriented bars that shift over time (*Figure 3c*). It is possible that this balance, between a code that is optimal for prediction and one that prioritizes efficient wiring, might underlie differences in the retina and LGN of different species. The retina of mice and rabbits contains many neurons with oriented and direction-tuned RFs, whereas cats and macaques mostly have center-surround RFs (*Scholl et al., 2013*). Efficient retinal wiring may be more important in some species, due, for example, to different constraints on the width of the optic nerve or different impacts of light scattering by superficial retinal cell layers.

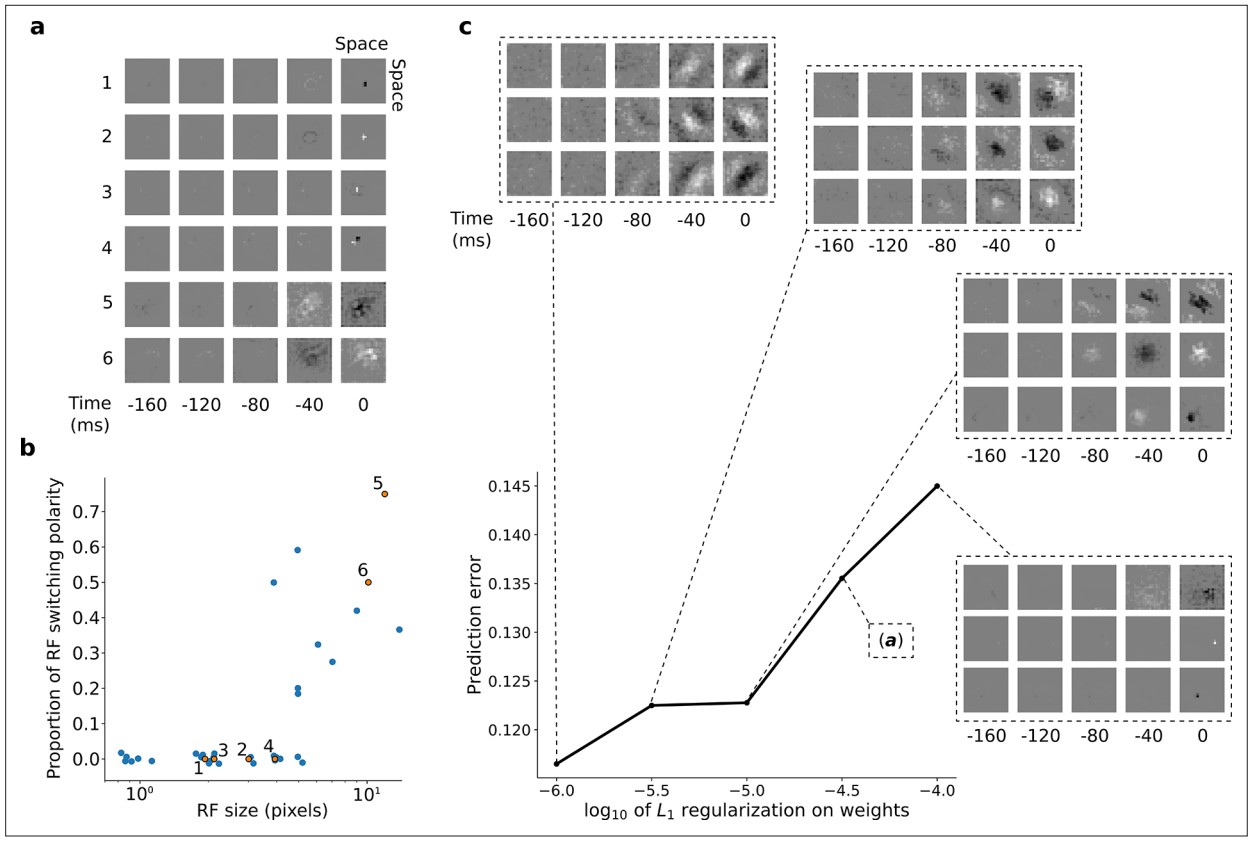

**Figure 3.** RFs of trained first stack of the model show retina-like tuning. (**a**) Example RFs with center-surround tuning characteristic of neurons in retina and LGN. RFs are small and do not switch polarity over time (units 1–4) or large and switch polarity (units 5–6), resembling cells along the parvocellular and magnocellular pathways, respectively. (**b**) RF size plotted against proportion of the weights (pixels) in the RF that switch polarity over the course of the most recent two timesteps. Units in **a** labeled and shown in orange. (**c**) Effect of changing regularization strength on the qualitative properties of RFs.

## Hierarchical temporal prediction produces simple and complex cell tuning

Using the trained center-surround-tuned network as the first stack, a second stack was added to the model and trained. The output of each second stack unit results from a linear-nonlinear-linear-nonlinear transformation of the visual input, and hence we estimated their RFs by reverse correlation with binary noise input. Many of the resulting RFs were Gabor-like over space (*Figure 4a and b*, I-II), resembling those of V1 simple cells (*Hubel and Wiesel, 1959*; *Hubel and Wiesel, 1965*; *Hubel and Wiesel, 1968*). The RF envelopes decayed into the past, and often showed spatial shifts or polarity changes over time, indicating direction or flicker sensitivity, as is also seen in V1 (*DeAngelis et al., 1993*; *Figure 4a and b*, I; *Figure 4—figure supplement 1*). Using full-field drifting sinusoidal gratings (*Figure 4a and b*, III-IV), we found that most units were selective for stimulus orientation, spatial and temporal frequency (*Figure 4a and b*, V-VII), and some were also direction selective (*Figure 4b*). Responses to the optimal grating typically oscillate over time between a maximum when the grating is in phase with the RF and 0 when the grating is out of phase (*Figure 4a and b*, IV). These response characteristics are typical of V1 simple cells (*Movshon et al., 1978a*).

In the third and fourth stack, we followed the same procedures as in the second stack. Most of these units are also tuned for orientation, spatial frequency, temporal frequency and in some cases for direction (*Figure 5a and b*, V-VII; *Figure 5—figure supplements 1 and 2*). However, while some units resembled simple cells, most resembled the complex cells of V1 and extrastriate visual areas (such as V2/V3/V4/MT; *Hubel and Wiesel, 1965*; *Hu et al., 2018*). Complex cells are tuned for orientation and spatial and temporal frequency, but are relatively invariant to the phase of the optimal grating *Movshon et al., 1978b*; each cell's response to its optimal grating has a high mean value and changes little with the grating's phase (*Figure 5a and b*, III-IV). Whether a neuron is assigned as simple or complex is typically based on the modulation ratio in such plots (<1 indicates complex; *Skottun et al., 1991*). Model units with low modulation ratios had little discernible structure in their RFs (*Figure 5a and b*, I-II), another characteristic feature of V1 complex cells (*Rust et al., 2005*; *Schwartz et al., 2006*).

We do not claim that any stack in the model corresponds directly to any specific stage along the visual pathway, rather that the order that tuning properties emerge as one moves up the stacks resembles the order that they emerge as one moves up the visual pathway. For example, it can be argued that at least some of the units in stack 4 may correspond to units seen in higher visual areas. With this caveat in mind, we quantified the tuning characteristics of units in stacks 2–4 and compared them to published V1 data (*Ringach et al., 2002*; *Figure 6a–j*). Simple cells have high modulation ratios and typically have RFs that can be well approximated by Gabor functions (*Jones and Palmer, 1987*; *Skottun et al., 1991*), while complex cells have low modulation ratios and typically have unstructured RFs (*Rust et al., 2005*; *Schwartz et al., 2006*). We included all putative simple cell-like units that responded to drifting gratings (model units with modulation ratios >1, whose RFs could be fitted by Gabors; *Figure 6—figure supplements 1 and 2*; see Methods) and all putative complex cell-like units that responded to drifting gratings (model units with modulation ratios <1) from stacks 2–4 in this analysis.

The distribution of modulation ratios is bimodal in both macaque monkey V1 (*Ringach et al., 2002*; *Skottun et al., 1991*) and our model (*Figure 6a*). Both model and real neurons were typically orientation selective, but with the model units having weaker tuning as measured by orientation bandwidth (median data [*Ringach et al., 2002*]: 23.5°, model: 37.5°; *Figure 6b*) and circular variance (median data [*Ringach et al., 2002*]: simple cells 0.45, complex cells 0.66; median model: simple cells 0.47, complex cells 0.85; *Figure 6c and d*). Orientation-tuned units (circular variance <0.9 or orientation bandwidth <150°) and direction-tuned units (direction selectivity >0.1) in the second stack were mostly simple (modulation ratios >1), whereas those in subsequent stacks became increasingly complex (*Figure 6a and c–f*). Circular variance varied with the modulation ratio in a similar way in both model (*Figures 6c–e*) and recorded V1 data (*Figure 6c–d and h*). A similar trend between model and data was also seen for orientation bandwidth and modulation ratio (*Figure 6f and i*).

Model units showed a range of direction selectivity preferences (*Figure 6g and j*) using a common direction selectivity index (DSI1, *Rochefort et al., 2011*). Simple cell-like units (60% with DSI1 ≥0.5; n=95; *Figure 6g*) tended to be more direction tuned than complex cell-like units (19% with DSI1 ≥0.5, n=205; *Figure 6g*), as is seen in cat V1 (*Kim and Freeman, 2016*). The distribution of direction

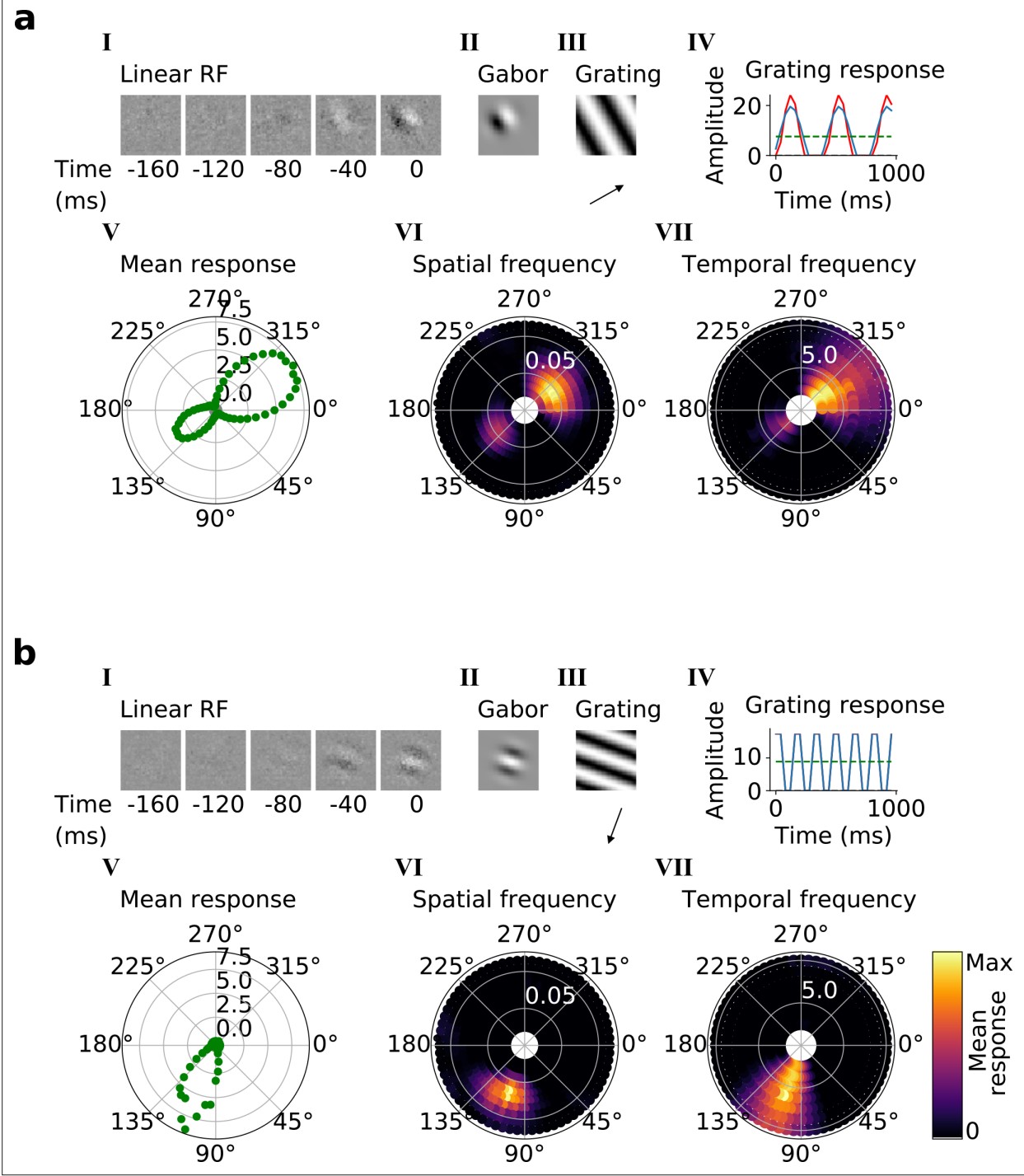

**Figure 4.** Qualitative tuning properties of example model units in stack 2. (**a, b**) Tuning properties of two example units from the 2nd stack of the model, including (I) the linear RF and (II) the Gabor fitted to the most recent time-step of the linear RF. (III) the drifting grating that best stimulates this unit. (IV) the amplitude of the unit's response to this grating over time. Red line: unit response; blue line: best-fitting sinusoid; gray dashed line: response to blank stimulus, which is zero and hence obscured by the x-axis. Note that the response (red line) is sometimes obscured by a perfectly overlapping best-fitting sinusoid (blue line). (V) The unit's mean response over time plotted against orientation (in degrees) for gratings presented at its optimal spatial and temporal frequency. (VI,VII) Tuning curves showing the joint distribution of responses to (VI) orientation (in degrees) and spatial frequency (in cycles/pixel) at the preferred temporal frequency and to (VII) orientation and temporal frequency (in Hz) at the preferred spatial frequency. In VI and VII the color represents the mean response over time to the grating presented. For more example units, see *Figure 4—figure supplement 1*.

The online version of this article includes the following figure supplement(s) for figure 4:

**Figure supplement 1.** Tuning properties of example units in stack 2.

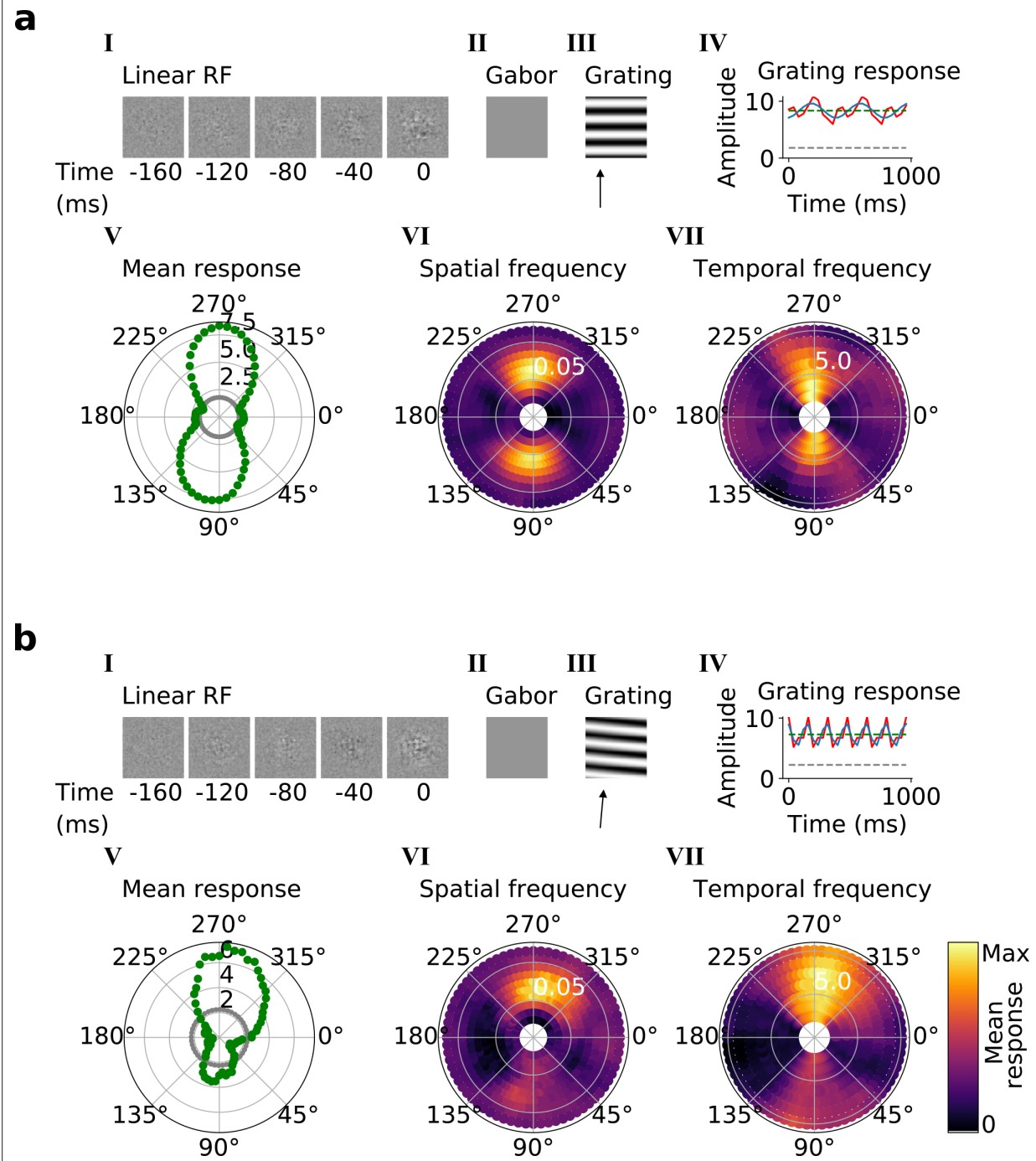

**Figure 5.** Qualitative tuning properties of example model units in stack 3. (**a, b**), As in *Figure 4a and b*. Note that **a**(II) and **b**(II) when blank indicate that a Gabor could not be well fitted to the RF. Also, note that in **a**(IV) and **b**(IV) the response to a blank stimulus is visible as a gray line, whereas in the corresponding plots in *Figure 4* this response is zero. For more example units, see *Figure 5—figure supplement 1*. Example units from stack 4 are shown in *Figure 5—figure supplement 2*.

The online version of this article includes the following figure supplement(s) for figure 5:

**Figure supplement 1.** Tuning properties of example units in stack 3.

**Figure supplement 2.** Tuning properties of example units in stack 4.

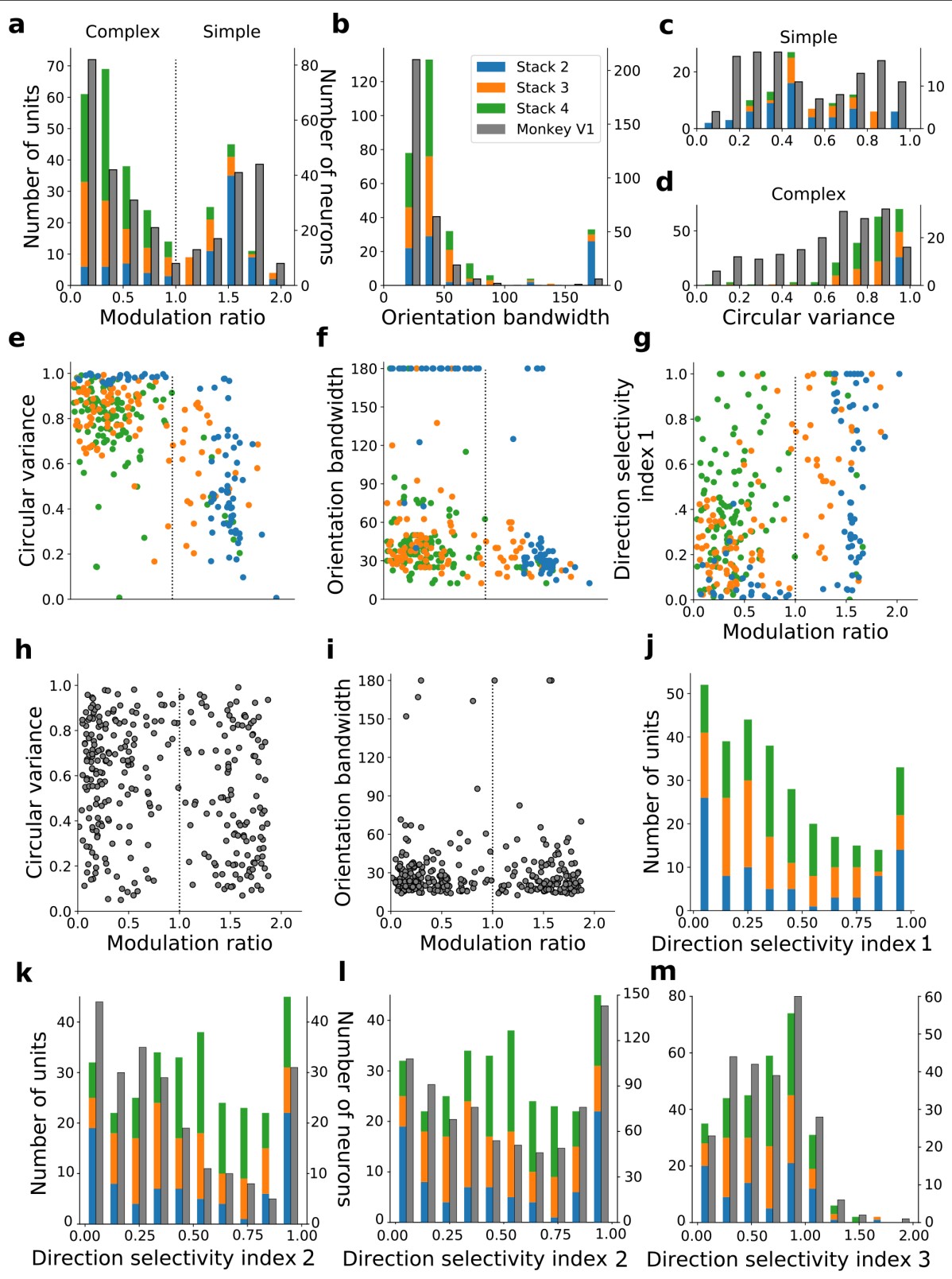

**Figure 6.** Quantitative tuning properties of model units in stacks 2–4 in response to drifting sinusoidal gratings and corresponding measures of macaque V1 neurons. (**a–d**) Histograms showing tuning properties of model and macaque V1 (**Ringach et al., 2002**) neurons as measured using drifting gratings. Units from stacks 2–4 are plotted on top of each other to also show the distribution over all stacks; this is because no stack can be uniquely assigned to V1. Modulation ratio measures how simple or complex a cell is; orientation bandwidth and circular variance are measures of orientation

*Figure 6 continued on next page*

*Figure 6 continued*

selectivity. (**e, f, g**) Joint distributions of tuning measures for model units. (**h, i**) Joint distributions of tuning measures for V1 data; note similarity to **e** and **f**. (**j**) Distribution of direction selectivity for model units (using direction selectivity index 1, see Methods). (**k,l**) Distribution of direction selectivity for model units and macaque V1 (**k** from *De Valois et al., 1982*; **l** from *Schiller et al., 1976*) using direction selectivity index 2. (**m**) Distribution of direction selectivity for model units and for cat V1 (from *Gizzi et al., 1990*) using direction selectivity index 3. In all cases, for units with modulation ratios >1, only units whose linear RFs could be well fitted by Gabors were included in the analysis. To see the spatial RFs of these units and their Gabor fits, see *Figure 6—figure supplement 1*, and for those units not well fitted by Gabors see *Figure 6—figure supplement 2*. Some units had very small responses to drifting gratings compared to other units in the population. Units whose mean response was <0.1% of the maximum for the population were excluded from analysis. To see the RFs and Gabor fits and quantitative properties of the units of the present-predicting and the weight-shuffled control models, see *Figure 6—figure supplements 3–6*.

The online version of this article includes the following figure supplement(s) for figure 6:

**Figure supplement 1.** Linear RFs of model units that could be well fitted by Gabors and corresponding Gabors.

**Figure supplement 2.** Linear RFs of model units that could not be well fitted by Gabors and corresponding Gabors.

**Figure supplement 3.** Linear RFs of model units and corresponding Gabors for stacked autoencoding (present-predicting) model.

**Figure supplement 4.** Quantitative tuning properties of stacked autoencoding (present-predicting) model units in stacks 2–4 in response to drifting sinusoidal gratings and corresponding measures of macaque V1 neurons.

**Figure supplement 5.** Linear RFs of model units and corresponding Gabors for the model with the input weights to each unit in the network shuffled across space and time.

**Figure supplement 6.** Quantitative tuning properties of the model with input weights to each unit shuffled across space and time for units in stacks 2–4 in response to drifting sinusoidal gratings and corresponding measures of macaque V1 neurons.

selectivity in the model units was dominated by peaks at each extreme, with cells tending to be highly direction selective or non-direction selective, although also with a slight central peak of moderately selective units (*Figure 6j*). A similar trend is seen for a related measure of directionality selectivity (DSI2), and this trend is also evident in macaque V1 data (*Schiller et al., 1976*; *De Valois et al., 1982*; *Figure 6k and l*) with peaks at the two extremes, but a less evident central peak. Using a third direction selectivity index (DSI3), which is more sensitive due to subtracting out the response to a null stimulus, a dominant peak of direction-selective neurons is seen in the model units (at a value of ~1) and some units are even seen to be inhibited by the direction orthogonal to the preferred direction (DSI3 values >1). This is also observed in cat V1 data (*Gizzi et al., 1990*; *Figure 6m*).

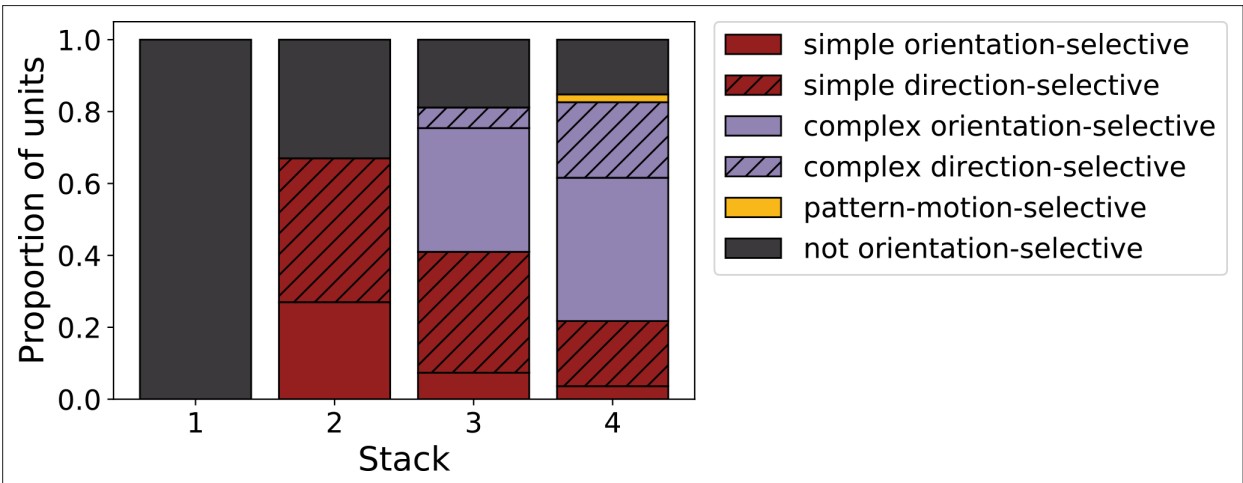

**Figure 7.** Progression of tuning properties across stacks of the hierarchical temporal prediction model. Proportion of simple (red), complex (purple) and non-orientation-selective (gray) units in each stack of the model as measured by responses to full-field drifting sinusoidal gratings. Simple cells are defined as those that can be well fitted by Gabors (pixel-wise correlation coefficient >0.4), are orientation-tuned (circular variance <0.9) and have a modulation ratio >1. Complex cells are defined as those that are orientation-tuned (circular variance <0.9) and have a modulation ratio <1. Crossed lines show number of direction-tuned simple (red) and complex (purple) units in each stack of the model. Direction-tuned units are simple and complex cells (as defined above) that additionally have a direction selectivity index >0.5. Also shown is the proportion of pattern-motion-selective units (yellow) as measured by drifting plaids.

We can summarize the kinds of units found as we progress from stack 1 to stack 4 by assigning each unit to a category. Taking all units that responded to drifting gratings, we defined orientation-selective simple-cell units as above (modulation ratio >1, Gabor fit correlation coefficient >0.4), but also with circular variance <0.9, and orientation-selective complex-cell units as above (modulation ratio <1), but also with circular variance <0.9; any remaining units are defined as non-orientation-selective (e.g. the center-surround units of stack 1). A clear progression from non-orientation-selective, to simple-cell-like, to complex-cell-like is seen in the units as one progresses up the stacks of the model (*Figure 7*). This bears some resemblance to the progression from the retina and LGN, which has few orientation-selective neurons, at least in cats and monkeys (*Barlow, 1953*; *Kuffler, 1953*; *Shapley and Perry, 1986*), to the geniculorecipient granular layer of V1, which tends to contain more simple cells, to the superficial layers of V1, where more complex cells have been found (*Ringach et al., 2002*; *Cloherty and Ibbotson, 2015*). This is also consistent with the substantial increase in the proportion of complex cells from V1 to extrastriate visual areas, such as V2 (*Cloherty and Ibbotson, 2015*).

## Some model units are tuned to two-dimensional features of visual motion

Simple and complex cells extract many dynamic features from natural scenes. However, their small bandpass RFs prevent individual neurons from tracking the motion of objects because of the aperture problem; the direction of motion of an edge is ambiguous, with only the component of motion

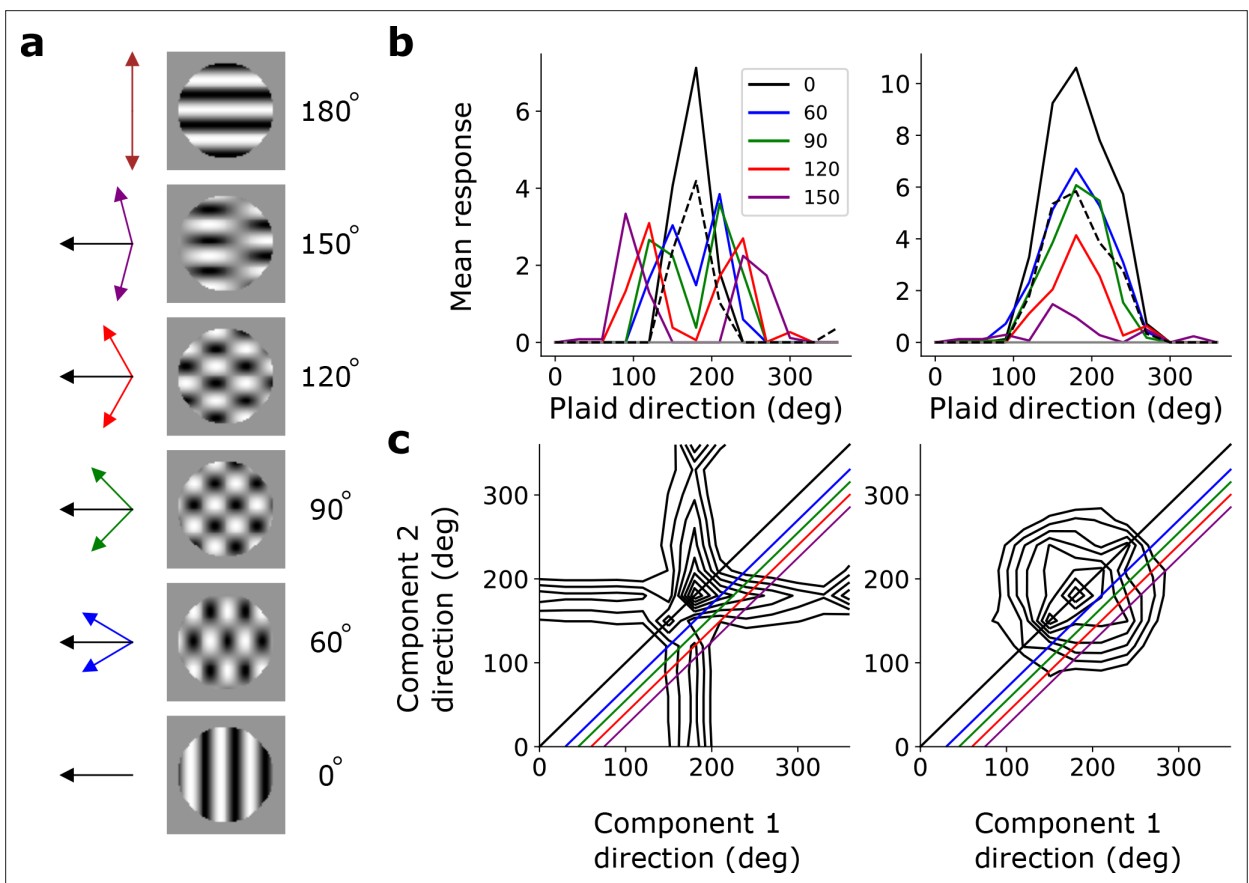

**Figure 8.** Pattern sensitivity. (**a**) Example plaid stimuli used to measure pattern selectivity. Black arrow, direction of pattern motion. Colored arrows, directions of component motion. (**b**) Direction tuning curves showing the response of an example component-motion-selective (left, from stack 2) and pattern-motion-selective (right, from stack 4) unit to grating and plaid stimuli. Colored lines, response to plaid stimuli composed of gratings with the indicated angle between them. Black solid line, unit's response to double intensity grating moving in the same direction as plaids. Black dashed line, response to single intensity grating moving in the same direction. Horizontal gray line, response to blank stimulus. (**c**) Surface contour plots showing response of units in **b** to plaids as a function of the direction of the grating components. Colored lines denote loci of plaids whose responses are shown in the same colors in **b**. Contour lines range from 20% of the maximum response to the maximum in steps of 10%. For clarity, all direction tuning curves are rotated so that the preferred direction of the response to the optimal grating is at 180°. Responses are mean amplitudes over time.

perpendicular to the cell's preferred orientation being represented. Pattern-selective neurons solve this problem, likely by integrating over input from many direction-selective V1 complex cells (*Movshon et al., 1985*; *Movshon and Newsome, 1996*; *Rust et al., 2006*; *Simoncelli and Heeger, 1998*; *Zeki, 1974*), and hence respond selectively to the motion of patterns as a whole. These neurons have been found in macaque extrastriate cortical area MT (*Movshon et al., 1985*; *Rodman and Albright, 1989*; *Rust et al., 2006*), cat anterior ectosylvian visual area (AEV) (*Scannell et al., 1996*), ferret posterior suprasylvian sulcus (PSS) (*Lempel and Nielsen, 2019*), and even in mouse V1 (*Palagina et al., 2017*).

To investigate pattern selectivity in our model units, we measured their responses to drifting plaids, comprising two superimposed drifting sinusoidal gratings with different orientations. The net direction of the plaid movement lies midway between these two orientations (*Figure 8a*). In V1 and MT, component-selective cells respond maximally when the plaid is oriented such that either one of its component gratings moves in the preferred direction of the cell (as measured by a drifting grating). This results in two peaks in plaid-direction tuning curves (*Movshon et al., 1985*; *Rust et al., 2006*; *Smith et al., 2005*). Conversely, pattern-selective cells have a single peak in their direction tuning curves, when the plaid's direction of movement aligns with the preferred direction of the cell (*Movshon et al., 1985*; *Rust et al., 2006*; *Smith et al., 2005*). We see examples of both component-selective units (in stacks 2–4) and pattern-selective units (only in stack 4) in our model, as indicated by plaid-direction tuning curves (*Figure 8b*) and plots of response as a function of the directions of each component (*Figure 8c*).

The hierarchical temporal prediction model is not explicitly designed to model the pathway in which MT lies (i.e. the dorsal pathway) over other regions of the visual pathway (such as those comprising the ventral visual stream). Some units in the higher stacks of the model could also represent features we have not explored here, but which are represented in V1 or other higher areas. However, in this study we have chosen to focus on motion sensitivity, as found in the dorsal pathway. Only a few (8%, n=3/37) of the direction-selective units in stack 4 are pattern-motion selective. This is a lower proportion than is seen in MT (~23%, n=179/792 [*Wang and Movshon, 2016*]), where the vast majority (>90%, [*Wang and Movshon, 2016*]) of neurons are direction sensitive (DSI >0.5). However, there are also direction-selective neurons in V1 and in higher cortical regions other than MT (e.g. V2, V3 [*Hu et al., 2018*]) that show little or no pattern-selectivity. Hence, if we consider the 4th stack to be not exclusively MT-like, but rather to comprise units that fall in all of these regions collectively, this will dilute the proportion of direction-selective units that are pattern-motion selective. We explore further possible reasons for the low proportion of pattern-motion selective units in the Discussion.

We also investigated what role the stimulus statistics play in the appearance of pattern-motion-selective units. Although the videos we use have a fair amount of camera motion in them, they also included clips with little or no visual motion. We introduced additional motion into the videos by creating versions that panned left or right at various speeds (see Methods). When we trained the temporal prediction model on this augmented dataset, we observed a substantial increase in the proportion of direction-selective units in all stacks, as well as an increase in the number of pattern-motion-selective units to 19, with some of them emerging in stacks 2 and 3 as well as stack 4. While there are phenomenological models (i.e. that are fitted to neural data) of pattern selectivity (*Rust et al., 2006*; *Simoncelli and Heeger, 1998*), to our knowledge, our model is the first normative model (i.e. where properties emerge from first principles applied to natural statistics) that produces this phenomenon.

## Determining the model properties critical for the units to resemble visual neurons

It is interesting to ask whether the tuning properties that emerge are due to the convolutional hierarchical structure of the model, the temporal prediction objective, or both together. At least some hierarchy or recurrency in model structure is required to obtain complex-cell-like units - the simple linear-nonlinear structure of hidden units in our previous non-hierarchical temporal prediction model is incapable of producing units resembling complex cells, although it can produce simple-cell-like units (*Singer et al., 2018*).

To investigate whether a model with the same convolutional hierarchical structure as our model, but with a different objective function than temporal prediction, could produce cells with tuning properties resembling those of visual neurons, we instead trained the model at each stack to efficiently

estimate the present input given present and past input, rather than to predict its future input. In other words, we trained each stack of this control model (which has some similarities to a stacked autoencoder, *Bengio et al., 2006*) to reproduce the most recent time step of its input from the same time step and the preceding 4 time-steps. We would expect such a model to have very limited capacity to represent the temporal structure of the movies. When we trained this model with the same form and dataset as the temporal prediction model, we indeed found that while the first stack could reproduce the center-surround spatial tuning seen in the retina, the model unit RFs had no temporal structure – they only depended on the present time step. Furthermore, this control model could not produce magnocellular-like units that switch polarity, nor could it produce units at any stack that resembled simple cells or complex cells by inspection, either spatially or temporally (*Figure 6—figure supplements 3 and 4*). By our definitions (*Figure 7*), there were only ~2% as many simple cells and no complex cells, compared to our standard temporal prediction model. Furthermore, none of the units in this control model showed pattern-motion-selectivity. This suggests that both hierarchy and temporal prediction acting together produce the diverse tuning properties we observe in our model.

As a further control model, we shuffled the input weights across space and time for each stack of the trained hierarchical temporal prediction model, and then probed the network with stimuli. In this case, the RFs in all stacks lacked discernable structure (*Figure 6—figure supplement 5*), and could not be fitted by Gabor functions, being neither center-surround nor simple-cell-like (only ~1% relative to the standard model met the criteria for simple-cell-like). To look for complex-cell-like units, we measured the model units' responses to drifting gratings (*Figure 6—figure supplement 6*). They showed only patchy selectivity for spatial and temporal frequency and orientation, and there were almost no (<1% relative to standard model) units responding like complex cells in any stack. In addition, none of the units showed pattern-motion-selectivity. The fact that we do not see center-surround, simple cells, complex cells or pattern-motion-selective cells shows that the precise ordering of the weights learned through temporal prediction (as opposed to these weights being randomly shuffled) is important for capturing the tuning phenomena observed in the brain.

Finally, we examined the importance of temporal structure in the stimuli for the results seen in the model. To test this, we shuffled the order of the frames in each clip, reducing their similarity over time. In the first stack, center-surround spatial tuning remained, but temporal structure of the RFs spanned the full five frames available and without temporal asymmetry or increased weighting near the present. By inspection, no RFs resembled simple cells, and by our definitions, simple-cell-like units and complex-cell-like units were much rarer across stacks (~4% and ~13% relative to the standard model). None of the units showed pattern-motion-selectivity.

## Predicting neural responses to natural stimuli

A standard method to assess a model's capacity to explain neural responses is to measure how well it can predict neural responses to natural stimuli. We did this for our model for two datasets. The first comprised single-unit neural responses recorded from awake macaque V1 to images of natural scenes (*Cadena et al., 2019*). The second, particularly relevant to our model, consisted of multi-unit responses from anesthetized macaque V1 to movies of natural scenes (*Nauhaus and Ringach, 2007*; *Ringach and Nauhaus, 2009*). Estimating neural responses to such dynamic natural stimuli has received less attention in models of visual processing.

We measured the capacity of the temporal prediction model to predict the neural responses of these datasets, and compared it to three other published models of the visual processing hierarchy. The first, the visual geometry group (VGG) model, is a deep convolutional neural network optimized for object recognition that has been trained using many labeled images (*Simonyan and Zisserman, 2014*). This supervised model has been commonly used for explaining responses in visual cortex (*Cadena et al., 2019*). The second, the Berkeley wavelet transform (BWT) model, is a descriptive model that encapsulates the classical approach to modeling simple and complex receptive fields by using Gabor-like wavelets (*Willmore et al., 2008*). The third model is the PredNet model (*Lotter et al., 2016*; *Lotter et al., 2020*), which is an unsupervised model trained to predict the immediate future frame of natural movies from recent past frames. This model differs from our temporal prediction approach in that it only applies this frame prediction at the first layer, with subsequent layers seeking an efficient representation of the activity in the lower layer.

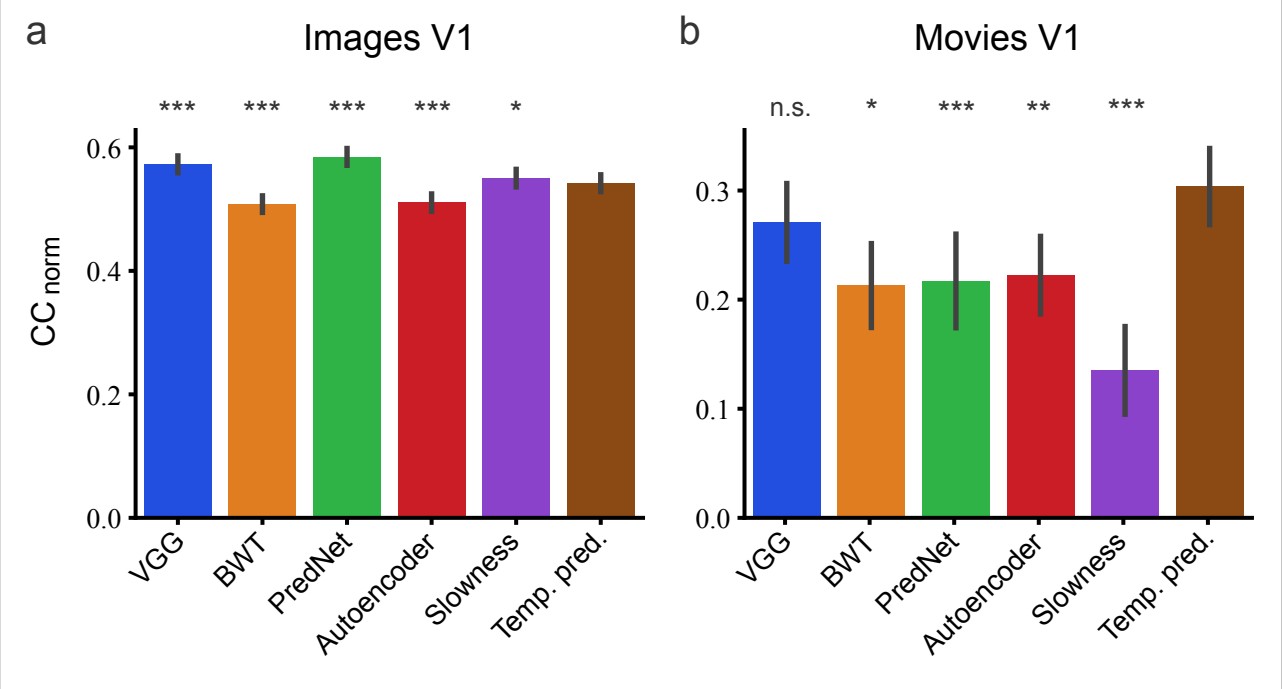

**Figure 9.** Assessment of different models in their capacity to predict neural responses in macaque V1 to natural stimuli. (**a**) Prediction performance for neural responses to images (*n* = 166 neurons). Neural data from *Cadena et al., 2019*. (**b**) *Prediction performance for neural responses to movies (n = 23 neurons). Neural data from* ***Nauhaus and Ringach, 2007*** *and* ***Ringach and Nauhaus, 2009***. The error bars are standard errors for the population of recorded neurons. Prediction performance that does not differ significantly by a bootstrap method (see Methods) from the temporal prediction model is marked by n.s., while significantly different CC_norm values are denoted by *p <0.05, **p <0.01, ***p <0.001.

Furthermore, we also modified our model so that comparison could be made with alternative unsupervised objective functions to temporal prediction. One variant of our model had the same architecture, but each stack was instead optimized to reproduce the current (rather than future) input along with a sparsity constraint on the activity (a sparse autoencoder model). Another variant also had the same architecture, but each stack was instead optimized to find slowly varying features while decorrelating activity across units (a slowness model). We also reduced the kernel spacing (stride) and made a few other small modifications (see Methods) to these two models and to the standard temporal prediction model to make them more directly comparable to the BWT, VGG and PredNet models, which use a very fine spacing of kernels in their first layer.

The method we adopted to assess the capacity of the different models to predict responses to natural stimuli has previously been used to assess models of recording data from macaque monkeys and mice (*Schrimpf et al., 2018*; *Storrs et al., 2021*; *Zhuang et al., 2021*; *Mineault et al., 2021*; *Conwell et al., 2021*; *Nayebi et al., 2023*). This involved taking a model's unit activity in a layer or stack (over a span of two time-steps, 66ms), performing dimensionality reduction on it to a standard size (500 dimensions), and then finding the best linear-nonlinear fit (using cross validation) from this activity to the neural responses over time. Finally, the model and the linear-nonlinear fit were used to predict neural responses to a held-out fraction of the natural stimuli, and the accuracy of this prediction was measured. The assessment measure that we used (CC_norm) was the Pearson correlation coefficient between the neural response and the model's prediction of the neural response, normalized to take into account intrinsic neural variability (*Schoppe et al., 2016*; *Hsu et al., 2004*).

On predicting the neural responses to images (*Figure 9A*), relative to the temporal prediction model, the BWT and autoencoder models performed slightly worse (bootstrap, n = 166, p <$10^{-4}$ and p <$10^{-4}$, respectively, see Methods) and the VGG, PredNet and slowness models performed slightly better (bootstrap, n = 166, p <$10^{-4}$, p <$10^{-4}$ and p = $5.3 \times 10^{-3}$, respectively). However, for the dynamic visual stimuli, the results were quite different (*Figure 9B*). The BWT, PredNet, autoencoder, and slowness models all performed worse than the temporal prediction model (bootstrap, n = 23, p = $1.6 \times 10^{-2}$, p = $2 \times 10^{-4}$, p = $2.5 \times 10^{-3}$ and p <$10^{-4}$, respectively), with only the VGG model being not

significantly different (bootstrap, n = 23, n.s.). Hence, the temporal prediction model, an unsupervised model trained on unlabeled data with no explicit constraints imposed by either neural or behavioral data, rivals a supervised model (VGG) trained using a labeled behavioral dataset in predicting neural responses in V1 to dynamic stimuli. Thus, the temporal prediction model is a reasonable contender as an unsupervised model of V1 responses to natural stimuli, particularly dynamic natural stimuli.

## Discussion

We have presented a simple model that hierarchically instantiates temporal prediction – the hypothesis that sensory neurons are optimized to efficiently predict the immediate future from the recent past. This model has several advantages: it is unsupervised, it operates over spatiotemporal inputs, and its hierarchical implementation is general, allowing the same network model to learn features resembling each level of the visual processing hierarchy with little fine-tuning or modification to the model structure at each stage. This simple model accounts for many spatial and temporal tuning properties of cells in the dorsal visual pathway, from the center-surround tuning of cells in the retina and LGN, to the spatiotemporal tuning and direction selectivity of V1 simple and complex cells, and to some degree the motion processing of pattern-selective cells. Furthermore, it can predict neural responses in V1 to movies of natural scenes to a level equal to or better than other models of visual processing, such as supervised image recognition models, predictive coding models, and hierarchical sparse or slow activity models.

### Relation to biology

Although this work suggests that temporal prediction may explain why many features of sensory neurons take the form that they do, we are agnostic as to whether the features are hard-wired by evolution or learned over the course of development. This is likely to depend on the region in question, with retina more hard-wired and cortex a mixture of innate tuning and learning from sensory input (*Huberman et al., 2008*; *Kiorpes, 2015*). If the predictive features are learned, this suggests that while some neurons represent these features, a fraction of neurons in the cortex (or elsewhere) might represent the prediction error used to train the network. Indeed, there is evidence that some cortical neurons may represent prediction error (*Auksztulewicz et al., 2023*; *Fiser et al., 2016*; *Rubin et al., 2016*).

Our model is trained using backpropagation over a single hidden layer for each stack. Although the biological plausibility of backpropagation has been questioned, increasingly biologically realistic methods for training networks are being developed (*Whittington and Bogacz, 2019*). One advantage of our model, for comparison to the biology, is that it does not require backpropagation across more than one hidden layer and isolates learning within each stack, reducing the need to find biologically plausible alternatives to backpropagation that scale with depth (*Bartunov et al., 2018*). This stands in contrast to deep supervised models.

There are a number of further developments that could be made to our model that may even better capture features of the biology, notably the inclusion of recurrent connections within and between layers or using spiking units. Incorporating recurrent processing may lead to an improvement in the match between the number of pattern-selective units in the model and the proportion of these units seen in MT, as this has been argued to be important by some (*Pack et al., 2001*), but not all (*Rust et al., 2006*) physiological studies of pattern-selectivity in MT. It could also help capture complex spatiotemporal tuning properties of other types of visual neurons, for example those exhibited by some retinal ganglion cells (*Meister and Berry, 1999*). Furthermore, the videos used to train our model typically involve animals moving in natural environments filmed with still or slowly panning cameras. Incorporating other types of motion into the training videos, such as self-motion or eye movements, may also produce a closer match between the response properties of our model units and those seen in the biology. Indeed, when additional panning was incorporated into the training videos, the proportion of pattern-motion-selective units in the model increased, providing an experimentally testable prediction from our model. Finally, although we propose that temporal prediction is an important method for extracting potentially useful features of natural inputs, additional constraints are likely to further refine those features that are of direct relevance to specific behavioral goals.

## Applications of the model

Algorithms involving prediction have a long history of use in engineering and signal processing (*de Jager, 1952*; *Kalman, 1960*; *Kalman and Bucy, 1961*). Our model also has potential applications for such use, for example, as an unsupervised learning algorithm to initialize deep networks for supervised tasks. In semi-supervised and transfer learning, models are trained using unlabeled data to produce useful representations that help in the performance of specific tasks (such as object classification etc.) and then fine-tuned using few labeled training examples. Sparse coding models, independent component analysis (ICA) and denoising autoencoders have been widely used in these paradigms (*Bengio et al., 2006*; *Bengio, 2012*; *Bengio et al., 2012*; *Hinton et al., 2006*; *Lee et al., 2009*; *Raina et al., 2007*). More recently, models performing next-frame video prediction have been used in an attempt to find representations that are useful for goals such as action recognition (*Srivastava et al., 2015*; *Vondrick et al., 2016*), object classification (*Canziani and Culurciello, 2017*; *Vondrick and Torralba, 2017*) and action-generation in video games (*Oh et al., 2015*). Very few studies (*Lotter et al., 2020*; *Palm, 2012*), however, have made any attempt to compare the representations learned by these models to those of sensory neurons in the brain.

## Predictions of the model

There are a number of biological predictions suggested by our model. One prediction common to all unsupervised models, which distinguishes them from supervised and reinforcement models, is that neurons in sensory cortex should learn stimulus feature selectivity that is dependent on stimulus statistics even in the absence of reinforcement signals. Some studies have shown that altering stimulus statistics can change cortical representations; for example, rearing cats with goggles that restrict the range of visible orientations influences the orientation tuning of V1 neurons (*Tanaka et al., 2006*). However, the animals could still be seeking rewards and avoiding punishments in such altered environments. To more securely test whether feature selectivity can be learned without reinforcement signals, an experiment would be required that separates reinforcement from stimulus statistics.

There are also a number of predictions that distinguish hierarchical temporal prediction from other unsupervised frameworks, such as predictive coding (*Rao and Ballard, 1999*). Although many possible variants of predictive coding exist (*Spratling, 2017*), by focusing on the framework of *Rao and Ballard, 1999*, we can draw distinctions from hierarchical temporal prediction. First, predictive coding suggests that the majority of feedforward projections between cortical regions should transmit prediction error, whereas temporal prediction suggests that these projections should transmit the presence of predictive features. This could be directly tested using physiological measurements.

Second, feedback connectivity is integral to perception in predictive coding, but not for temporal prediction. Indeed, many aspects of visual perception can occur so rapidly that there may not be time for feedback to operate (*Fabre-Thorpe, 2011*; *Thorpe et al., 1996*), supportive of predominantly feed-forward models such as the one we present here. This does, however, raise the question of what role feedback connections would play in the temporal prediction framework. We would suggest that they may be important for contextually modulating neuronal selectivity (*Crist et al., 2001*; *Gilbert et al., 2001*), perhaps using higher level statistics to help predict lower level statistics. Model differences could be tested by optogenetically targeting cortico-cortical feedback connections (*Nurminen et al., 2018*), and then investigating the behavioural consequences.

Third, temporal correlational structure in sensory input is extremely important for the temporal prediction model, but not for predictive coding, which can learn from still images. This implies that an animal reared in a visual environment, perhaps a virtual environment using goggles, where the frame order of natural visual inputs is randomly shuffled, should develop altered cortical tuning, not just in the temporal domain, but also in the spatial domain. If no changes were seen this would argue against temporal prediction as a learning mechanism. A recent study (*Matteucci and Zoccolan, 2020*) reported that rearing rats in a box with frame-scrambled natural movies played upon the walls led to a substantial degradation and reduction in complex cells, consistent with the importance of temporal prediction, at least for those cells. No alterations in the simple cell population were found, but this may be because much of simple cell tuning is hard-wired before eye opening (*Huberman et al., 2008*). Hence simple cell spatiotemporal tuning in rats may still be optimized for temporal prediction, but by evolutionary processes rather than learning processes in the animal's lifetime. However, the movies used in this study were shown at a lower frame rate than rats' flicker fusion frequency, and

some aspects of their visual inputs were not frame-scrambled (screen edges and the rat's own body when still and under self-motion), implying that they did experience some temporal continuity. This may account for the lack of changes in simple cells and the residual fraction of complex cells. Further studies of this sort in different species will help settle these questions.

## Comparison to other normative models

Normative models of visual processing derive visual neuron tuning properties from natural scene statistics and optimization for one or more normative principle. As we have mentioned, there are several normative models of visual processing, based on a range of principles, which can account for a number of properties of visual neurons. These theories include predictive coding, sparse coding, independent component analysis (ICA) and temporal coherence.

The predictive coding framework postulates that sensory systems learn the statistical regularities present in natural inputs, feeding forward the errors caused by deviations from these regularities to higher areas (*Huang and Rao, 2011*; *Rao and Ballard, 1999*; *Srinivasan et al., 1982*). In this process, the predictable components of the input signal are subtracted and the prediction errors are fed forward through the hierarchy. This should be distinguished from temporal prediction, which performs selective coding, where predictive elements of the input are explicitly represented in neuronal responses and fed forward, and non-predictable elements are discarded (*Bialek et al., 2001*; *Chalk et al., 2018*; *Salisbury and Palmer, 2016*; *Figure 1*).

Sparse coding (*Olshausen and Field, 1997*; *Olshausen and Field, 1996*), which shares similarities with predictive coding (*Huang and Rao, 2011*), is built on the idea that an overcomplete set of neurons is optimized to represent inputs as accurately as possible using only a few active neurons for a given input. ICA (*Bell and Sejnowski, 1997*; *van Hateren and van der Schaaf, 1998*) is a related framework that finds maximally independent features of the inputs. Sparse coding and ICA are practically identical in cases where a critically complete code is used. In these frameworks, as with predictive coding, the aim is to encode all current or past input, whereas in temporal prediction, only features that are predictive of the future are encoded and other features are discarded.

Another set of approaches, slow feature analysis (*Wiskott and Sejnowski, 2002*) (SFA) and slow subspace analysis (*Kayser et al., 2001*), stem from the idea of temporal coherence (*Földiák, 1991*), which suggests that a key goal of sensory processing is to identify slowly varying features of natural inputs. SFA is closely related to temporal prediction because features that vary slowly are likely to be predictive of the future. However, SFA and temporal prediction may give different weighting to the features that they find (*Creutzig and Sprekeler, 2008*), and SFA could also fail to capture features that do not vary slowly, but are predictive of the future. One notable study suggests that the features found by SFA can be comparably predictable over time to those features found by networks trained to find predictable features (*Weghenkel and Wiskott, 2018*). However, predictable features are not the same as predictive features. For example, if a horizontally oriented feature is always followed by a vertically oriented feature in the next frame, but the horizontally oriented features are randomly scattered over time, then the horizontally oriented features, while being predictive of the future, are not themselves predictable.

In the following, we focus on unsupervised normative models trained on natural inputs, because they are the most relevant to our model. Broadly, these models can be divided into several categories: local models, trained to represent features of a specific subset of neurons, such as simple cells in V1, and hierarchical models, which attempt to explain features of more than one cell type (such as simple and complex cells) in a single model. These two categories can be further divided into models that are trained on natural spatial inputs (images) and those that are trained on natural spatiotemporal inputs (movies).

## Local models of V1 simple or complex cells

Among local models, sparse coding and ICA are the standard normative models of V1 simple cell RFs (*Bell and Sejnowski, 1997*; *Olshausen, 2002*; *Olshausen and Field, 1997*; *Olshausen and Field, 1996*; *van Hateren and Ruderman, 1998*; *van Hateren and van der Schaaf, 1998*). Typically, these models are trained using still natural images and have had remarkable success in accounting for the spatial features of V1 simple cell RFs (*Bell and Sejnowski, 1997*; *Olshausen and Field, 1997*; *Olshausen and Field, 1996*; *van Hateren and van der Schaaf, 1998*). However, models trained on

static images are unable to account for the temporal aspects of neuronal RFs, such as direction selectivity or two-dimensional motion processing.

The ICA and sparse coding frameworks have been extended to model features of spatiotemporal inputs (*Chalk et al., 2018*; *Olshausen, 2002*; *van Hateren and Ruderman, 1998*). While these models capture many of the spatial tuning properties of simple cells, they tend to produce symmetric temporal envelopes that do not match the asymmetric envelopes of real neurons (*Singer et al., 2018*). Capturing temporal features is especially important when building a normative model of the dorsal visual stream, which is responsible for processing cues related to visual motion.

When trained to find slowly varying features in natural video inputs, SFA models (*Berkes and Wiskott, 2005*) find features with tuning properties that closely resemble those of V1 complex cells, including phase invariance to drifting sinusoidal gratings and end- and side-inhibition. A sparse prior must be applied to the activities of the model units in order to produce spatial localization – a key feature of V1 complex cells (*Lies et al., 2014*). Although SFA can account for complex cell tuning, on its own this framework does not provide a normative explanation for simple cells.

## Hierarchical models trained on natural images

Predictive coding (*Rao and Ballard, 1999*) provides a powerful framework for learning hierarchical structure from visual inputs in an unsupervised learning paradigm. When applied to natural images, predictive coding has been used to explain some nonlinear tuning properties of neurons in V1, such as end-stopping (*Rao and Ballard, 1999*), and, when constrained to have sparse responses, reproduces orientation tuning that resembles the spatial tuning of simple cells. However, it is not clear whether this framework can reproduce the phase-invariant tuning of complex cells, nor has it been shown to account for direction selectivity or pattern-motion sensitivity (*Movshon et al., 1985*).

Hierarchical ICA models (and related models) provide another approach. These consist of two-layer networks that are trained on natural images with an independence prior placed on the unit activities (*Hyvärinen and Hoyer, 2000*; *Hyvärinen and Hoyer, 2001*; *Karklin and Lewicki, 2009*; *Osindero et al., 2006*). They have been shown to produce simple-cell-like subunits in the first layer of the network and phase-invariant tuning reminiscent of complex cells in the second layer. However, these models typically incorporate aspects that specifically encourage complex cell-like characteristics in the form of a quadratic nonlinearity resembling the complex cell energy model (*Adelson and Bergen, 1985*). In some models, phase invariance is enforced by requiring the outputs of individual subunits to be uncorrelated (*Hyvärinen and Hoyer, 2000*; *Körding et al., 2004*). This is in contrast to our model where the phase invariance is learned as a consequence of the general principle of finding features that can efficiently predict future input. An advantage to learning the invariance is that the identical model architecture can then be applied hierarchically to explain features in higher visual areas without changing the form of the model.

## Hierarchical models trained on natural spatiotemporal inputs

Some hierarchical models based on temporal coherence have been trained on spatiotemporal inputs (videos of natural scenes). These models have been shown to capture properties of both simple and complex cells (*Hurri and Hyvärinen, 2003*; *Kayser et al., 2001*; *Körding et al., 2004*). Typically, they share a similar structure and assumptions with the hierarchical ICA models outlined above, consisting of a two-layer model where the outputs of one or more simple cell subunits are squared and passed forward to a complex cell layer. Other models have combined sparsity/independence priors with a temporal slowness constraint in a hierarchical model (*Berkes et al., 2009*; *Cadieu and Olshausen, 2012*; *Chen et al., 2018*; *Hyvärinen et al., 2003*). Since sparsity constraints tend to produce simple cell tuning and slowness constraints result in complex cell tuning, these models produce units with both types of selectivity. This contrasts with our model, which produces both types of selectivity with a single objective and also accounts for some tuning features of higher visual areas.

Recently, a hierarchical predictive coding model (PredNet [*Lotter et al., 2020*; *Lotter et al., 2016*]) has been developed, which combines features of Rao and Ballard's (*Rao and Ballard, 1999*) predictive coding model and temporal prediction (*Figure 1—figure supplement 1*). This shares some similarities with our hierarchical temporal prediction model, but also has several important differences and is substantially more complex (*Figure 1—figure supplement 1*). Similar to our temporal prediction model, the PredNet model tries to predict the future input, rather than compress current input like

classical predictive coding. However, a notable and substantial difference from our model is that in PredNet, as in predictive coding, it is the prediction error, rather than the predictive signal contained in the hidden unit activity, that is passed up as input to the next layer of the model. PredNet captures some interesting physiological or psychophysical aspects of visual processing, such as end-stopping, sequence learning, illusory contours and the flash-lag effect (*Lotter et al., 2020*). However, it has not been demonstrated that PredNet captures the phenomena that we describe, namely the parvocellular/magnocellular division, orientation- and direction-tuned simple and complex cells, and pattern-direction selectivity.

## Quantitative comparison to different models

Because the visual system has evolved to process natural stimuli, an important consideration in whether a computational model captures visual processing is whether it can predict neural responses to such stimuli. To this end, we estimated how well the hierarchical temporal prediction model can predict responses in awake macaque V1 to movies and images of natural scenes. Of particular interest are responses to movies, as our model is focused on the neural processing of dynamic stimuli. We compared the temporal prediction model to other models of visual system organization. Two of these models have identical structure to the temporal prediction model, but implement different underlying principles; one model estimates the current rather than future input alongside a sparse activity constraint (sparse coding principle) and the other finds slowly-varying features (slowness principle). We also made comparisons with three other key models of visual processing, the PredNet model, based on the principle of predictive coding, a wavelet model (BWT), inspired by the classic Gabor model, and a deep supervised model trained on image recognition (VGG).

It is important to note that it can be difficult to determine whether the success or failure of a given model to predict neural responses is a consequence of the key principles underlying the model or the details of its implementation. Making different models comparable in the way they are implemented is a challenge, and while we have striven to do this, there is always scope for further improvement. Furthermore, there are also other classes of models that could be examined (e.g. *Wang et al., 2021*; *Wang et al., 2018*), training datasets that could be used (e.g. http://moments.csail.mit.edu, https://cmd.rcc.uchicago.edu, https://deepmind.com/research/open-source/kinetics), or neural datasets to which the models could be compared (e.g. a comparison to the primate ventral pathway data as in *Khaligh-Razavi and Kriegeskorte, 2014*; *Yamins et al., 2014*; *Zhuang et al., 2021*).

Thus, our analysis should not be seen as definitively ruling out the principles underlying any of the other models, but rather an attempt to ascertain whether temporal prediction is a good contender for an underlying principle. It is also important to point out that while the sparsity, slowness and PredNet models are, like temporal prediction, unsupervised normative models, the wavelet model is a descriptive model, and the VGG model is a supervised model fitted to human behavioral data (labeled images). The behavioral data provide quantitative measurements of the outputs of the visual system that constrain the VGG model, which is not the case for the other models in this comparison. Thus, in some senses, the VGG model is a fitted descriptive encoding model, rather than a normative unsupervised model.

We found that the performance of the hierarchical temporal prediction model at predicting the responses of V1 neurons to images was not substantially different from that of leading models, such as the VGG model. Importantly, for the V1 responses to movies, the temporal prediction model outperformed all the other models, except for the VGG model which it performed as well as. However, the VGG model has the advantage of labeled data, and was also trained on far more data than our model, whereas the temporal prediction model is unsupervised. Our findings therefore indicate that temporal prediction remains a leading contender as an underlying normative unsupervised principle governing the neural processing of natural visual scenes.

## Conclusions

Previous studies using principles related to temporal prediction have produced non-hierarchical models of the retina or primary sensory cortex and demonstrated retina-like RFs (*Ocko et al., 2018*), simple-cell like RFs (*Chalk et al., 2018*; *Singer et al., 2018*), or RFs resembling those found in primary auditory cortex (*Singer et al., 2018*). However, any general principle of neural representation should be extendable to a hierarchical form and not tailored to the region it is attempting to explain. Here,

we show that temporal prediction can indeed be made hierarchical and that it reproduces the major motion-tuning properties that emerge along the visual pathway from retina and LGN to V1 granular layers to V1 superficial layers and potentially some properties of the extrastriate dorsal cortical pathway. This model captures not only linear tuning features, such as those seen in center-surround retinal neurons and direction-selective simple cells, but also nonlinear features seen in complex cells, with some indication that it may also reproduce properties of the pattern-sensitive neurons that are found in area MT. The capacity of our hierarchical temporal prediction model to account for many tuning features at several levels of the visual system suggests that the same framework may well explain many more features of the brain than we have so far investigated. Iterated application of temporal prediction, as performed by our model, could be used to make predictions about the tuning properties of neurons in brain pathways that are much less well understood than those of the visual system. Our results suggest that, by learning behaviorally useful features from dynamic unlabeled data, temporal prediction may represent a fundamental coding principle in the brain.

## Methods
### Data used for model training and testing
#### Visual inputs
Videos (grayscale, without sound, sampled at 25 frames-per-second) of wildlife in natural settings were used to create visual stimuli for training the artificial neural network. The videos were obtained from http://www.arkive.org/species and contributed by: BBC Natural History Unit, http://www.gettyimages.co.uk/footage/bbcmotiongallery; BBC Natural History Unit & Discovery Communications Inc, http://www.bbcmotiongallery.com; Granada Wild, http://www.itnsource.com; Mark Deeble & Victoria Stone Flat Dog Productions Ltd., http://www.deeblestone.com; Getty Images, http://www.gettyimages.com; National Geographic Digital Motion, http://www.ngdigitalmotion.com. The videos consisted of animals moving in natural environments, filmed with panning, still and occasionally zooming cameras. The longest dimension of each video frame was clipped to form a square image. Each frame was then down-sampled (using bilinear interpolation) over space, to provide 181x181 pixel frames. The video patches were cut into non-overlapping clips, each of 20 frames duration (800ms). We used a training set of $N$ = ~1305 clips from around 17 min of video, and a validation set of $N$ = ~145 clips. Finally, each clip was normalized by subtracting the mean and dividing by the standard deviation of that clip.

#### Adding panning to videos
To investigate the effect of using videos with additional motion, we separately trained the model on an augmented dataset where additional panning was incorporated into each of the videos. To do this, we shifted each frame horizontally by a given number of pixels left or right relative to the previous frame. The dataset that included panning was comprised of the original videos, plus these videos with each of left and right panning of 1, 2, and 3 pixels per frame. The results of training the model on the dataset with additional panning are mentioned in relation to patternmotion selectivity, but it should be noted that the main results presented are from the model when trained on the original dataset.

### Hierarchical temporal prediction model
#### The basic model and cost function
First, we will describe the basic hierarchical temporal prediction model without convolution (*Figure 1*). Although this model is not used in this paper, it is the basis for the convolutional hierarchical temporal prediction model presented here. The basic temporal prediction model consists of stacked feedforward single-hidden-layer neural networks. The first stack receives as input the movie, and predicts future frames of the movie from a number of past frames. The second stack receives as input the activities of the hidden units of the first stack, and predicts the future activity of the first stack from a number of time steps of past activity of the first stack. The third stack receives as input the activities of the second stack, and so on.

 Each stack is optimized independently, starting with the lowest stack and moving up the stacks, the output of the trained lowest stack providing input for the next stack which was then trained, and so on. The cost function of each stack, minimized by backpropagation, is the sum of the squared error between the prediction and the target, plus an $L_1$-norm regularization term, which is proportional to

the sum of absolute values of all weights in the network. This regularization tends to drive redundant weights to near zero and provides a parsimonious network. Note, a single stack for this model is almost exactly the same as the non-hierarchical model from *Singer et al., 2018*.

## The convolutional model and cost function

The convolutional hierarchical temporal prediction model used in the present study (*Figure 2*) is the same as the basic hierarchical temporal prediction model outlined above, except that the networks involved are convolutional neural networks. Convolutional networks recognize that neural tuning for similar sets of spatially-restricted features is likely to be found at every point in visual space, due to the relatively local and translation-invariant statistics of natural scenes. Hence, rather than learn the feature separately for every point in space, convolutional networks ensure that the same set of features is learned at every point in space by using convolution, aiding and accelerating learning. Formulating the model as convolutions of tensors provides a compact notation.

The convolutional hierarchical temporal prediction model consists of stacked feedforward single-hidden-layer 3D convolutional neural networks. Each stack consists of an input layer, a convolutional hidden layer and a convolutional output layer. Note, there are no sub-sampling layers. Each hidden layer consists of a number of convolutional hidden units, each convolutional hidden unit being the set of hidden units tuned to the same feature at every point in space. Each convolutional hidden unit (over time and 2D space; *Figure 2*) performs 3D convolution over its inputs using its particular convolutional kernels, and its output is determined by passing the result of this operation through a rectified linear function. Thus, hidden units have non-negative activity. Following the hidden layer there is a convolutional output layer, which performs linear convolution. Each stack is trained to minimize the difference between its output layer and its target. The target is the input at the immediate future time-step.

The first stack of the model is trained to predict the immediate future frame (40ms) of unfiltered natural video inputs from the previous 5 frames (200ms). This 5-frame span was chosen after earlier exploration of the trained model indicated that most of the non-negligible weights tended to fall within this span. Each subsequent stack is then trained to predict the immediate future hidden-unit activity of the stack below it from the past hidden-unit activity in response to the natural video inputs. This process is repeated until 4 stacks have been trained. The first stack uses 50 convolutional hidden units and this number is doubled with each added stack, until we have 400 units in the 4th stack.

More formally, each stack of the model can be described by a network of the same form. Throughout this section on the convolutional model, bold capitalized variables are rank-3 tensors over time and 2D-space, otherwise variables are scalars. The network has an input layer $i = 1$ to $I$ input channels, a single hidden layer of $j = 1$ to $J$ convolutional hidden units, and an output layer of $k = 1$ to $K = I$ output channels.

For input channel $i$, the input $\mathbf{U}_i$ is a rank-3 tensor spanning time and 2D-space with $x = 1$ to $X$ and $y = 1$ to $Y$ spatial positions, and $t = 1$ to $T$ time steps. Each input channel $\mathbf{U}_i$ is the output $\mathbf{H}_j$ of convolutional hidden unit $j$ in the stack below (with $i = j$), except for the first stack which has only a single input channel ($i = 1$), which is the grayscale video. Hence, each subsequent stack has as many input channels ($I$) as the number of convolutional hidden units (feature maps) in the previous stack.

For a given convolutional hidden unit $j$, the output of the unit is a rank-3 tensor over time and 2D-space, $\mathbf{H}_j$, and is given by,

$$\mathbf{H}_j = f\left( b_j + \sum_{i=1}^{I} \mathbf{U}_i * \mathbf{W}_{ji} \right) \tag{1}$$

where $\mathbf{W}_{ji}$ is the input weights kernel between each input channel $i$ and hidden unit $j$, $b_j$ is the bias of hidden unit $j$, and f() is the rectified linear function. The operator $*$ is the 3D convolutional operator over the two spatial and one temporal dimensions of the input. Each hidden layer kernel $\mathbf{W}_{ji}$ is 3D with size $(X', Y', T')$. No zero padding is applied to the input.

The output of the network predicts the future activity of the input. Hence, the number of input channels ($I$) always equals the number of output channels ($K$) for each stack. The activity $\hat{\mathbf{V}}_k$ of each output channel $k$ is the estimate of the true future $\mathbf{V}_k$ given the past $\mathbf{U}_i$, where $\mathbf{V}_k$ is simply $\mathbf{U}_i$ shifted one time-step into the future, and $k = i$. This prediction $\hat{\mathbf{V}}_k$ is estimated from the hidden unit output $\mathbf{H}_j$ by

$$\hat{\mathbf{V}}_k = c_k + \sum_{j=1}^{J} \mathbf{H}_j * \mathbf{M}_{kj} \tag{2}$$

The parameters in *Equation 2* are the output weight kernels $\mathbf{M}_{kj}$ (the weights between each hidden unit $j$ and output channel $k$) and the bias $c_k$. Each output kernel $\mathbf{M}_{kj}$ is 3D with size $(X', Y', 1)$, and predicts a single time-step into the future based on hidden layer activity in that portion of space.

There is a slight exception for the first stack, where to cope with the size of the images the spatial stride value of the convolution in *Equation 1* was greater than one (it was 10). To ensure that the predicted output provided by *Equation 2* has the same size as the input when a stride of >1 is used, the hidden layer representation is first dilated over by adding $\mathbf{s} - 1$ zeros between adjacent input elements, where $\mathbf{s} = (s_1 = 10, s_2 = 10, s_3 = 1) = (\text{space, space, time})$ is the stride of the convolutional operator in the hidden layer. This dilation makes *Equation 2* into what is known as a fractionally-strided convolutional layer (*Dumoulin and Visin, 2018*).

Each stack is optimized independently, starting with the lowest stack and moving up the stacks, the hidden unit activity $\mathbf{H}_j$ of the trained lowest stack providing the input for the next stack, which is then trained, and so on. For each stack, the parameters of every $\mathbf{W}_{ji}$, $\mathbf{M}_{kj}$, $b_j$, and $c_k$ were optimized for the training set by minimizing the cost function given by,

$$E = \frac{1}{NKXYT} \sum_{n=1}^{N} \sum_{k=1}^{K} \left\| \hat{\mathbf{V}}_{kn} - \mathbf{V}_{kn} \right\|_2^2 + \lambda \left( \sum_{i=1}^{I} \sum_{j=1}^{J} \left\| \mathbf{W}_{ji} \right\|_1 + \sum_{j=1}^{J} \sum_{k=1}^{K} \left\| \mathbf{M}_{kj} \right\|_1 \right) \tag{3}$$

where $n$ indicates the video clip number and $\|\|_p$ is the entrywise $p$-norm of the tensor over time and 2D-space, where $p = 2$ provides the square root of the sum of squares of all values in the tensor and $p = 1$ provides the sum of absolute values. Thus, $E$ is the sum of the squared error between the prediction $\hat{\mathbf{V}}_{kn}$ and the target $\mathbf{V}_{kn}$, plus an $L_1$-norm regularization term, which is proportional to the sum of absolute values of all weights in the network and its strength is determined by the hyper-parameter $\lambda$.

The resulting set of stacked networks can then operate as a single feed-forward convolutional network, with *Equation 1* providing the transformation at each stack, using the weights and biases learned for that stage. *Equations 2 and 3* are simply used for training the stacks. In *Figure 2*, we use a bracketed superscript to indicate the weight matrices associated with each stack; stack 1, $\mathbf{W}_{ji}^{(1)}$, stack 2, $\mathbf{W}_{ji}^{(2)}$, stack 3, $\mathbf{W}_{ji}^{(3)}$, and stack 4, $\mathbf{W}_{ji}^{(4)}$.

## Implementation details

The networks were implemented in Python (https://lasagne.readthedocs.io/en/latest/; http://deep-learning.net/software/theano/). The objective function for each stack was minimized using backpropagation as performed by the Adam optimization method (*Kingma and Ba, 2014*) with hyperparameters $\beta_1$ and $\beta_2$ kept at their default settings of 0.9 and 0.999, respectively, and the learning rate ($\alpha$) varied as detailed below. Training examples were split into minibatches of 32 training examples each.

During model network training, several hyperparameters were varied, including the regularization strength ($\lambda$) and the learning rate ($\alpha$). For each hyperparameter setting, the training algorithm was run for 1000 iterations. The effect of varying $\lambda$ on the prediction error (the first term of *Equation 3*) and receptive field structure of the first stack is shown in *Figure 3*. For all subsequent stacks, we varied $\lambda$ between $10^{-5}$ and $10^{-7}$ and picked the network with the lowest prediction error (mean squared error) on a held-out validation set. We measured the predictive capacity of each network by taking the average prediction error on the validation set over the final 50 iterations. We also explored the time span of frames into the past and set it sufficiently long (5 frames) that in the trained model there was typically very little weighting for the frame furthest into the past. The settings for each stack are presented in *Table 1*.

## Model unit spatiotemporal extent and receptive fields

Due to the convolutional form of the hidden layer, each hidden unit can potentially receive from a certain span over space and time. We call this the unit's spatial and temporal extent. For stack 1, this this extent is given by the kernel size (21x21 x 5, space x space x time). For stack 2, the extent of each hidden unit is a function of its kernel size and the kernel size and stride of the hidden units in the previous stack, resulting in an extent of 41x41 x 9. Similarly, the extent of each hidden unit in stack 3

is 61x61 x 13 and in stack 4 is 81x81 x 17. The RF size of a unit can be considerably smaller than the hidden unit's extent if weights assume a value close to zero.

In the first stack of the model, the combination of linear weights and nonlinear activation function are similar to the basic linear non-linear (LN) model (*Chichilnisky, 2001*; *Simoncelli et al., 2004*) commonly used to describe neuronal RFs. Hence, the input weights between the input layer and a hidden unit of the model network are taken directly to represent the unit's linear RF, indicating the features of the input that are important to that unit. The output activities of hidden units in stacks 2–4 are transformations with multiple linear and nonlinear stages, and hence we estimated their linear RFs by applying reverse correlation to 100,000 responses to binary noise input with amplitude ±3 to stack 1.

## Fitting Gabors to model unit receptive fields

We fitted 2-dimensional Gabor functions to the most recent time-step of each of model unit's linear RF. The Gabor function has been shown to provide a good approximation for most spatial aspects of simple cell visual RFs (*Jones and Palmer, 1987*). The 2D Gabor is given as:

$$G\left(x', y'\right) = A \cdot \exp\left(-\left(\frac{x'}{\sqrt{2}\sigma_x}\right)^2 - \left(\frac{y'}{\sqrt{2}\sigma_y}\right)^2\right) \cdot \cos\left(2\pi f x' + \phi\right) \tag{4}$$

where the spatial coordinates $(x', y')$ are given by the following transformation of the RF center $(x_0, y_0)$ with rotation of the RF by its spatial orientation $\theta$:

$$x' = (x - x_0) \cdot \cos\theta + (y - y_0) \cdot \sin\theta \tag{5}$$

$$y' = -\left(x - x_0\right) \cdot \sin\theta + (y - y_0) \cdot \cos\theta \tag{6}$$

$\sigma_x$ and $\sigma_y$ provide the width of the Gaussian envelope in the $x'$ and $y'$ directions, while $f$ and $\phi$ parameterize the spatial frequency and phase of the sinusoid along the $x'$ axis. $A$ parameterizes the height of the Gaussian envelope. For each RF, the parameters $(x_0, y_0, \sigma_x, \sigma_y, \theta, f, \phi)$ of the Gabor were fitted by minimizing the mean squared error between the Gabor model and the RF using Python code modified from: https://github.com/JesseLivezey/gabor_fit, copy archived at *Livezey, 2019*.

Units that could be well fitted by Gabor functions, that is, where the pixel-wise correlation coefficient between the model unit RF and the fitted Gabor was >0.4, were included as putative simple cells (*Figure 6—figure supplements 1 and 2*).

## In vivo V1 receptive field data

Responses to drifting gratings measured using recordings from V1 simple and complex cells were compared against the model (*Figure 6*). The in vivo data were taken from *Ringach et al., 2002*.

## Receptive field size and polarity

We measured the size of the RFs of the units in the first stack and examined the relationship between the RF size and the proportion of the RFs switching polarity. For each unit, all RF pixels (weights) in the most recent time-step with absolute values ≥50% of the maximum absolute value in that time-step are included in the RF. The RF size was determined by counting the number of pixels fitting this criterion. We then counted the proportion of pixels included in the RF that changed sign (either positive to negative or vice versa) between the two most recent timesteps. The relationship between these two properties for the units in the first stack is shown in *Figure 3b*.

## Drifting sinusoidal gratings

In order to characterize the tuning properties of the model's visual RFs, we measured the responses of each unit to full-field drifting sinusoidal gratings. For each unit, we measured the response to gratings with a wide range of orientations, spatial frequencies and temporal frequencies until we found the parameters that maximally stimulated that unit (giving rise to the highest mean response over time). We define this as the optimal grating for that unit. In cases where orientation or tuning curves were measured, the gratings with optimal spatial and temporal frequency for that unit were used and were varied over orientation. Each grating alternated between an amplitude of ± 3 on a gray (0)

background. Some units, especially in higher stacks, had weak or no responses to drifting sinusoidal gratings. To account for this, we excluded any units with a mean response (over time) of <1% of the maximum mean response of all the units in that stack. As a result of this, 17/100, 99/200 and 286/400 units were excluded from the 2nd, 3rd, and 4th stacks, respectively.

We measured several aspects of the V1 neuron and model unit responses to the drifting gratings. For each unit, we measured the circular variance, orientation bandwidth, modulation ratio and direction selectivity.

## Circular variance

Circular variance (CV) is a global measure of orientation selectivity. For a unit with mean response-over-time $r_q$ to a grating with angle $\theta_q$, with angles spanning the range of 0–360° in equally spaced intervals of 5° and measured in radians, the circular variance is defined as (***Ringach et al., 2002***):

$$CV = 1 - \frac{\sqrt{(\Sigma_q r_q \sin(2\theta_q))^2 + (\Sigma_q r_q \cos(2\theta_q))^2}}{\Sigma_q r_q} \tag{7}$$

## Orientation bandwidth

We also measured the orientation bandwidth (***Ringach et al., 2002***), which provides a more local measure of orientation selectivity. First, we smoothed the direction tuning curve with a Hanning window filter with a half-width at half-height of 13.5°. We then determined the peak of the orientation tuning curve. The orientation angles closest to the peak for which the response was $1/\sqrt{2}$ (or 70.7%) of the peak response were measured. The orientation bandwidth was defined as half of the difference between these two angles. We limited the maximum orientation bandwidth to 180°.

## Modulation ratio

We measured the modulation ratio of each unit's response to its optimal sinusoidal grating. The modulation ratio is defined as:

$$MR = F_1/F_0 \tag{8}$$

where $F_1$ is the amplitude of the best-fitting sinusoid to the unit's response over time to the drifting grating. $F_0$ is the mean response to the grating over time.

## Direction selectivity index

To measure the direction selectivity index, we obtained each unit's direction tuning curve at its optimal spatial and temporal frequency. We measured the peak of the direction tuning curve, indicating the unit's response to gratings presented in the preferred direction ($r_p$), the response to the grating presented in the opposite (non-preferred) direction ($r_{np}$), and the response to a blank gray image ($r_{null}$). We examined three different direction selectivity indexes based on measures that have been used in neurophysiological studies. Direction selectivity index 1 (DSI1) (for example, in ***Rochefort et al., 2011***) is defined as:

$$DSI1 = (r_p - r_{np})/(r_p + r_{np}) \tag{9}$$

Direction selectivity index 2 is defined as:

$$DSI2 = (r_p - r_{np})/r_p \tag{10}$$

This is a simple transformation of the measures used by ***Schiller et al., 1976*** and ***De Valois et al., 1982***. The measure in ***Schiller et al., 1976*** is calculated as $100 r_{np}/r_p$; to convert their data to DSI2 we divided by 100 and then subtracted the values from one. Another difference between the DSI2 measure we used with the model and the measure of direction selectivity in ***Schiller et al., 1976*** is that they took the average responses to moving light and dark bars, whereas we used drifting gratings. ***De Valois et al., 1982*** used $r_{np}/r_p$, so to convert their data to DSI2 we subtracted their measure from one. We performed these transformations to provide a measure that goes from 0 to 1 to be more consistent with the other direction selectivity indexes.

Direction selection index 3 (as used, for example in ***Gizzi et al., 1990***) is defined as:

$$DSI3 = 1 - (r_{np} - r_{null})/(r_p - r_{null}) \qquad (11)$$

The direction tuning data were extracted from the relevant papers using WebPlotDigitizer (https://automeris.io/WebPlotDigitizer).

## Drifting plaid stimuli

In order to test whether units were pattern selective, we measured their responses to drifting plaid stimuli. Each plaid stimulus was composed of two superimposed half-intensity (amplitude 1.5) sinusoidal gratings with different orientations. The net direction of the plaid movement lies midway between these two orientations (*Figure 8a*). Plaids with a variety of orientations, spatial frequencies, temporal frequencies and spatial extents (as defined by the extent of a circular mask) were tested. For each unit, the direction tuning curves of the optimal plaid stimulus (that giving rise to the largest mean response over time) were measured (*Figure 8b and c*).

## Quantitative model comparison for neural response prediction

We compared how well the temporal prediction model and five other models could predict neural responses. The models that we compared to the temporal prediction model were the following: the visual geometry group (VGG) model – a spatial deep convolutional model trained in a supervised manner on labeled data for image recognition, which has been shown to predict well neural responses in V1 (*Cadena et al., 2019*); the Berkeley wavelet transform model – a descriptive spatial model of V1, and three unsupervised spatiotemporal models – a leading predictive coding model (PredNet), a slowness model and an autoencoder model with sparse activity. The models were compared for their capacity to predict neural responses to images and movies of natural scenes.

### V1 neural response datasets

We used two publicly available neural response datasets recorded in V1 of awake non-human primates. The first dataset contains recordings in response to natural images and the second dataset contains recordings in response to natural videos. For simplicity and to avoid confusion with model units, we will refer to single-unit and multi-unit recordings simply as 'neurons'. For those recordings, all stimuli were presented in grayscale with the animals trained to fixate on a small target. For both datasets, we used 80% of the data as a cross-validation set for training and setting hyperparameters (validation) and the remaining 20% of the data was held out for testing the models. For the videos, we split the dataset by video files rather than slicing videos in time. We also normalized all data by subtracting the mean and dividing by the standard deviation of each respective cross-validation dataset.

### Spatial V1 dataset

The image dataset contains the responses of 166 single-unit recordings in V1 to 7250 natural images (repeated 4 times) (*Cadena et al., 2019*). The natural images were from the ImageNet dataset (*Russakovsky et al., 2015*), with some further augmented using a textural synthesis model (*Gatys et al., 2015*; *Cadena et al., 2019*) (to further increase higher-order correlations within the dataset). We resampled the images to 112×112 pixels as input to the models.

The electrophysiological recordings came from two awake adult male macaque monkeys (*Macaca mulatta*) using a 32-channel linear silicon probe (approved by the Baylor College of Medicine Institutional Animal Care and Use Committee). The images were presented for 60ms (covering 2 degrees of visual angle) with spike counts extracted in the window 40–100ms after image onset. Following spike sorting, the dataset comprised 166 single-unit recordings that were driven by the visual stimuli (with at least 15% of the total variance accounted for from the stimulus). For a more in-depth description of the dataset acquisition see *Cadena et al., 2019*.

### Spatiotemporal V1 dataset

The video dataset comprised 23 multi-unit recordings in V1 in response to 10 videos of 30s duration each (repeated 10 times) (*Nauhaus and Ringach, 2007*; *Ringach and Nauhaus, 2009*). The videos were snippets from four different movies (Sleeper, Benji, Goldfinger and Shakespeare in Love). We resampled all spatial dimensions to 112×112 pixels as input to the models.

Electrophysiological recordings were obtained from anesthetized adult male macaque monkeys (*Macaca fascicularis*) using a 10×10 extracellular electrode array (approved by the UCLA Animal Research Committee). Videos were displayed at 30 frames-per-second (covering 2–6 degrees of visual angle) with spike counts binned in 33.33ms intervals. For a more in-depth description of the dataset acquisition see *Ringach and Nauhaus, 2009*.

## Models

### Visual Geometry Group (VGG)

The VGG model is a large convolutional neural network (CNN) with 16 convolutional and 3 fully connected layers (*Simonyan and Zisserman, 2014*) trained on the large ImageNet dataset containing more than a million images categorized into 1000 classes. As done by *Cadena et al., 2019*, we used the activation maps in layer 6 of the CNN model to predict the real neural responses.

### Berkeley Wavelet Transform (BWT)

The Berkeley wavelet transform (BWT, *Willmore et al., 2008*) is a static wavelet model that is inspired by the Gabor-like tuning (*Jones and Palmer, 1987*) of simple cells in V1. It performs a multiscale wavelet decomposition of each input image. Like Gabor filters, the wavelets are tuned in position, orientation and spatial frequency, and they can be divided into even and odd pairs with similar tuning. Unlike Gabor filters, they also form a complete orthonormal basis, making the decomposition inexpensive to compute. The sum of the square of the activities of corresponding pairs is used to model complex cells. Due to the wavelets being scaled by a power of 3, we zero-padded the spatial input dimensions of the input data to the closest power of 3. We used the simple- and complex-like units together to predict the neural responses.

### PredNet

The PredNet model is a multi-layered network, optimized to predict future frames from past frames in a movie (*Lotter et al., 2016*; *Lotter et al., 2020*). We used the pre-trained model from these publications, which was trained using the KITTI dataset (sampled at 10 frames per second) comprising footage from a roof-mounted camera on a car driving around a city (*Geiger et al., 2013*). We recognize that the PredNet model has been trained on movies with a different frame rate than the movies that were used as stimuli for the neural recordings, but even movies with identical frame rates can have different temporal characteristics, depending on their content. We therefore decided to use PredNet as optimized by the authors, rather than a potentially less well optimized version of PredNet retrained by us. The PredNet model is inspired by predictive coding theory (*Rao and Ballard, 1999*), reformulated into a modern deep learning context (i.e. built with convolutional and long short-term memory (LSTM) layers and optimized using backpropagation). As stipulated by predictive coding, the model has feedforward units that transmit the errors between predictions and lower-level activity and feedback units that transmit representations of the predictions of lower-level activity. As suggested by *Lotter et al., 2020*, for each stack (layer), we used the feedforward error representation units to fit to the real neural responses, as these units have been described to be most brain-like.

### Modifications to the temporal prediction model

We made certain modifications to the temporal prediction model for the neural response prediction tasks. First, to be more consistent with the comparison models, we employed more units within the first stack (100 instead of 50) and used a smaller convolutional stride (4 instead of 10). Second, we employed the softplus activation function (*Zheng et al., 2015*) (with $\beta = 1$ and threshold = 20) instead of the rectified linear activation function for all units. Softplus is similar to a rectified linear function, but has a smooth rather than abrupt change in slope. Third, we predicted the input at time $t + 2$ rather than at time $t + 1$. Fourth, for speed of training, instead of training on a stack-to-stack basis, we trained all stacks at once, and detached gradients from flowing between the stacks (to emulate the stack-to-stack training). We used the Adam optimizer with default parameters and trained the model for 300 epochs with a learning rate of $10^{-4}$. A value of $\lambda = 10^{-4}$ was used for stack one. Unless noted otherwise, all other hyperparameters were the same (*Table 1*).

Finally, we used a different and more diverse natural stimulus dataset for training (which we also used to train the slowness and autoencoder with activity sparsity models). We recorded this dataset

ourselves using an iPhone 7, and it consisted of 32 diverse movies (each of 9.6s duration sampled at 120 frames-per-second) of everyday objects and scenes (e.g. a fish swimming, a dog walking, a ball rolling, and trees moving in the wind) and different recording techniques (e.g. visual flow from forward motion, panning, still and moving content). We spatially downsampled the recordings from 720×1280 pixels to 144×256 pixels using the bilinear method (and then cropped to 140×240 pixels) and temporally downsampled the recordings from 120 frames-per-second to 30 frames-per-second by taking every 4th frame. We trained the model using batches of 15 clips of 30 frames in length (corresponding to approximately 1s of stimulus). We also employed data augmentation (*Taylor and Nitschke, 2018*) to artificially double the training dataset, by having a 50% chance of left-right flipping each clip when each batch is loaded.

## Stacked slowness model

The slow feature analysis (SFA) model extracts slowly-varying features from an input signal (*Wiskott and Sejnowski, 2002*; *Weghenkel and Wiskott, 2018*). We implemented a variant hereof using the same architecture of the modified temporal prediction model (*Equation 1*, four hierarchical stacks) with the cost function replaced by one with a slowness objective. Each stack finds the slowly-varying features of the hidden unit activity of the stack below (or the value of the stimuli for the first stack). There is no *Equation 2* in this model. This new cost function of each stack minimizes the mean-squared difference between consecutive time-steps of model unit activity, subject to activity decorrelation and sparse weight regularization.

$$E = \frac{1}{NJX_{\mathrm{H}}Y_{\mathrm{H}}\left(T_{\mathrm{H}}-1\right)} \sum_{n=1}^{N}\sum_{j=1}^{J} \left\|\dot{\mathbf{H}}_{jn}\right\|_2^2 + \frac{2}{NJ\left(J-1\right)} \sum_{n=1}^{N}\sum_{j=1}^{J}\sum_{j'=j+1}^{J} \left\|\rho_{njj'}\right\|_1 + \lambda \left(\sum_{i=1}^{I}\sum_{j=1}^{J} \left\|\mathbf{W}_{ji}\right\|_1\right) \quad (12)$$

Here, $\dot{\mathbf{H}}_{jn}$ is a rank-3 tensor over 2D-space (of width $X_{\mathrm{H}}$ and height $Y_{\mathrm{H}}$) and time (of duration $T_{\mathrm{H}}$). It is the difference in activity between consecutive time steps in the tensor $\mathbf{H}_{jn}$ of model hidden unit activity, for channel $j$ and video clip $n$. As stipulated within the SFA algorithm, we applied an additional constraint of decorrelating hidden unit activity, by minimizing the Pearson correlation coefficient $\rho_{nji}$ between the activity-over-time of the $j^{\mathrm{th}}$ and $j'^{\mathrm{th}}$ channel over the $n^{\mathrm{th}}$ clip. Before calculating the correlation coefficients, we added a small random variable (drawn from a uniform distribution between 0 and $10^{-4}$) to the activity of each hidden unit to avoid the numeric instability of division by zero. Lastly, the activity of every hidden unit was normalized using batch normalization (*Ioffe and Szegedy, 2015*). Again, we used the Adam optimizer with default parameters to train the model and we did so for 300 epochs with a learning rate of $10^{-4}$. Unlike the temporal prediction model, we cannot use the prediction performance to set $\lambda$, so instead we chose the value of $\lambda$ (from those explored) that gave the receptive fields that most closely resembled those of real neurons. Unless noted otherwise, all other hyperparameters were the same as *Table 1* and all other aspects of the model and its training were the same as the modified temporal prediction model described above.

## Stacked autoencoder with activity sparsity

As another control model, we trained a model in which each stack estimates the input at the current time step $t$ (rather than at the future time step $t+1$) of the spatiotemporal input signal, and with a sparse penalty on the hidden unit activity as well as on the weights. Again, we employed the same architecture (*Equation 1*, four hierarchical stacks) as the modified temporal prediction model. Each stack estimates the current activity of the stack below (or the value of the stimuli for the first stack). *Equation 2* also remains the same, but as it now estimates $\mathbf{U}_{kn}$ (which is the input $\mathbf{U}_{in}$ simply indexed with $k$ rather than $i$), the output of the equation becomes $\hat{\mathbf{U}}_{kn}$ instead of $\hat{\mathbf{V}}_{kn}$. We trained each stack with the following cost function.

$$E = \frac{1}{NKXYT} \sum_{n=1}^{N}\sum_{k=1}^{K} \left\|\hat{\mathbf{U}}_{kn}-\mathbf{U}_{kn}\right\|_2^2 + \lambda_1 \frac{1}{NJX_{\mathrm{H}}Y_{\mathrm{H}}T_{\mathrm{H}}} \sum_{n=1}^{N}\sum_{j=1}^{J} \left\|\mathbf{H}_{jn}\right\|_1 + \lambda_2 \left(\sum_{i=1}^{I}\sum_{j=1}^{J} \left\|\mathbf{W}_{ji}\right\|_1 + \sum_{j=1}^{J}\sum_{k=1}^{K} \left\|\mathbf{M}_{kj}\right\|_1\right) \quad (13)$$

Again, we used the Adam optimizer with default parameters to train the model and we did so for 300 epochs with a learning rate of $10^{-4}$. Unlike the temporal prediction model, we cannot use the

prediction capacity to set $\lambda_1$ and $\lambda_2$ , so instead we chose the values (from those values explored) that gave the receptive fields that most closely resembled those of real neurons. Unless noted otherwise, all other hyperparameters were the same as *Table 1* and all other aspects of the model and its training were the same as the modified temporal prediction model described above.

## Measuring the capacity of the models to predict neural responses to natural stimuli

For the spatial or spatiotemporal neural datasets, we ran the stimuli of the cross-validation set through each of the above models, obtaining the activity of their hidden units. One slight complication is that the spatial models (VGG and BWT) are designed to work with images, whereas the spatiotemporal models (all other models) are designed to work with videos. To pass the spatiotemporal stimuli (videos) into the spatial models (VGG and BWT), we fed in each frame individually as an image and concatenated the resulting unit activity to provide a time-series of activity. To pass the spatial stimuli (images) into the spatiotemporal models, we used a spatiotemporal sequence of repeated identical input frames.

For each model, or stack within a model, we estimated the neural responses by fitting a linear-nonlinear mapping from the latent unit activity (the first 500 principal components of the unit activity of the cross-validation set) of that model or stack. The minimum neural response latency from retina to V1 in macaque monkey is approximately 30ms, and there is a considerable spread of latencies (*Nowak et al., 1995*). To account for these transduction and conductance latencies we computed the latent unit activity $\mathbf{h}_t^{\mathrm{lat}}$ using the model unit activity (over space and channels) at time $t-1$ and $t-2$, which is a latency span of approximately 33–99ms. We used this latent unit activity to predict neural responses at time $t$. We used the principal components instead of directly fitting neural responses from unit activity as this standardizes the comparison between all the models, keeps the fitting procedure manageable (*Schrimpf et al., 2018*), and circumvents overfitting (*Storrs et al., 2021*). This technique of fitting on dimensionality-reduced unit activity has also been used in other studies (*Schrimpf et al., 2018*; *Zhuang et al., 2021*; *Mineault et al., 2021*; *Conwell et al., 2021*).

More formally, for each neuron $l$ in the dataset of $L$ neurons, we fitted the following linear-nonlinear readout, mapping latent model unit activity to the responses of the neuron:

$$\hat{r}_{lt} = \mathrm{softplus}\left(\mathbf{m}_l^{\mathsf{T}}\mathbf{h}_t^{\mathrm{lat}} + d_l\right) \tag{14}$$

Here, $\mathbf{m}_l$ is the neuron's readout weight vector, $d_l$ is a bias, and we used the softplus function as the non-linearity. We fitted the readouts by minimizing the negative Poisson likelihood of the model predictions $\hat{r}_{lt}$ and the neurons' responses $r_{lt}$ , and employed a L$_1$ penalty on the readout weights to avoid overfitting (*Cadena et al., 2019*).

$$E = -\frac{1}{LT}\sum_{l,t}\left(r_{lt}\ln\left(\hat{r}_{lt}\right) - \hat{r}_{lt}\right) + \lambda_{\mathrm{read}}\sum_l \|\mathbf{m}_l\|_1 \tag{15}$$

We used the Adam optimizer (with default parameters) for fitting, using a batch size of 1024, a learning rate of 0.002, and training for 300 epochs.

To do the readout fitting, we divided the neural response cross-validation set (whether for images or videos) and the corresponding model latent activity into a training set (80%) and a validation set (20%). We fitted the readout using the training set. Then, for validation, we ran the validation-set latent activity through the fitted readout to predict the corresponding neural responses, and measured the normalized correlation coefficient between the readout predictions and the neural responses (see Performance metrics). The above fitting and validation process was repeated 5 times, once for each distinct validation set. This fivefold cross-validation process was applied for 9 different log-spaced values of $\lambda_{\mathrm{read}}$ (ranging from $10^{-6.5}$ to $10^{-2.5}$). We then fitted the readout model on all the cross-validation data using the $\lambda_{\mathrm{read}}$ value that maximized the mean normalized correlation coefficient across all validation folds. The final performance was calculated on the test set, averaged over all neurons. For the unsupervised models (PredNet, slowness, sparseness, and temporal prediction), we reported the stack that performed the best. For a fairer comparison, we also allowed for the possibility of model units being tuned to different spatial sizes than the neurons. Thus, for all models, we applied

the above process at three spatial re-scalings of the model input (0.66×, 1× and 1.5×) and reported the performance for each model at its optimal scale.

## Performance metrics
### Normalized correlation coefficient

We used the normalized correlation coefficient to quantify all model fits (*Hsu et al., 2004*; *Schoppe et al., 2016*). The normalized correlation coefficient $CC_{norm}$ quantifies a model's performance as the Pearson correlation coefficient between the neural response and the model's prediction, divided by the maximum achievable correlation coefficient given neural noise. This measure is related to signal power explained (*Sahani and Linden, 2003*; *Shimazaki and Shinomoto, 2007*). A $CC_{norm}$ of zero indicates no correlation between the model and the neural response, while a $CC_{norm}$ of one indicates they are as correlated as possible given neural noise.

### Statistical tests

Statistical comparison of the different models was performed using a bootstrapping method (*Zhuang et al., 2021*). To calculate the probability, p, of model A performing better/worse statistically than model B, we sampled prediction scores over the neurons (paired, with replacement) for each model and calculated their respective means. We did this 10,000 times, counting the number of times model A had a larger mean than model B. Finally, this count was divided by 10,000 to obtain the probability, p, of model A having a larger mean than model B, with 1 - p being the probability of model B having a larger mean than model A.

## Acknowledgements

Yosef Singer was supported by the University of Oxford Clarendon Fund, the Oppenheimer Memorial Trust and the Goodger and Schorstein Research Scholarship in Medical Sciences. Luke Taylor was funded by the University of Oxford Clarendon Fund. Andrew King, Ben Willmore and Nicol Harper were supported by the Wellcome Trust (WT108369/Z/2015/Z).

## Additional information

### Funding

| Funder | Grant reference number | Author |
| --- | --- | --- |
| Wellcome Trust | WT108369/Z/2015/Z | Andrew J King<br>Nicol S Harper |
| University of Oxford<br>Clarendon Fund | | Yosef Singer<br>Luke Taylor |

The funders had no role in study design, data collection and interpretation, or the decision to submit the work for publication. For the purpose of Open Access, the authors have applied a CC BY public copyright license to any Author Accepted Manuscript version arising from this submission.

### Author contributions

Yosef Singer, Software, Investigation, Visualization, Methodology, Writing – original draft, Writing – review and editing, Formal analysis; Luke Taylor, Data curation, Software, Formal analysis, Validation, Investigation, Visualization, Methodology, Writing – review and editing; Ben DB Willmore, Supervision, Methodology, Writing – review and editing, Formal analysis; Andrew J King, Resources, Supervision, Funding acquisition, Writing – review and editing, Project administration; Nicol S Harper, Conceptualization, Supervision, Visualization, Methodology, Writing – review and editing, Formal analysis

## Author ORCIDs
Yosef Singer (iD) https://orcid.org/0000-0002-4480-0574
Luke Taylor (iD) http://orcid.org/0000-0002-4023-4958
Ben DB Willmore (iD) https://orcid.org/0000-0002-2969-7572
Andrew J King (iD) http://orcid.org/0000-0001-5180-7179
Nicol S Harper (iD) https://orcid.org/0000-0002-7851-4840

## Decision letter and Author response
Decision letter https://doi.org/10.7554/eLife.52599.sa1
Author response https://doi.org/10.7554/eLife.52599.sa2

## Additional files

### Supplementary files
• Transparent reporting form

### Data availability
All custom code used in this study was implemented in Python. The code for the models and analyses shown in Figures 1-8 and associated sections can be found at https://bitbucket.org/ox-ang/hierarchical_temporal_prediction/src/master/, (copy archived at *Singer et al., 2023a*). The V1 neural response data (*Ringach et al., 2002*) used for comparison with the temporal prediction model in Figure 6 came from http://ringachlab.net/ ("Data & Code", "Orientation tuning in Macaque V1"). The V1 image response data used to test the models included in Figure 9 were downloaded with permission from https://github.com/sacadena/Cadena2019PlosCB (*Cadena et al., 2019*). The V1 movie response data used to test these models were collected in the Laboratory of Dario Ringach at UCLA and downloaded from https://crcns.org/data-sets/vc/pvc-1 (*Nauhaus and Ringach, 2007*; *Ringach and Nauhaus, 2009*). The code for the models and analyses shown in Figure 9 and the associated section can be found at https://github.com/webstorms/StackTP (copy archived at *Singer et al., 2023b*) and https://github.com/webstorms/NeuralPred (copy archived at *Singer et al., 2023c*). The movies used for training the models in Figure 9 are available at https://figshare.com/articles/dataset/Natural_movies/24265498.

The following datasets were generated:

| Author(s) | Year | Dataset title | Dataset URL | Database and Identifier |
|---|---|---|---|---|
| Singer Y, Taylor L, Willmore BDB, King AJ, Harper NS | 2023 | Hierarchical temporal prediction captures motion processing along the visual pathway. Ox-ang/ hierarchical_temporal_ prediction | https://bitbucket.org/ ox-ang/hierarchical_ temporal_prediction/ src/master/ | Bitbucket, ox-ang/ hierarchical_temporal_ prediction/src/master/ |
| Singer Y, Taylor L, Willmore BDB, King AJ, Harper NS | 2023 | Hierarchical temporal prediction captures motion processing along the visual pathway. Webstorms/ StackTP | https://github.com/ webstorms/StackTP | GitHub, StackTP |
| Singer Y, Taylor L, Willmore BDB, King AJ, Harper NS | 2023 | Hierarchical temporal prediction captures motion processing along the visual pathway. Webstorms/ NeuralPred | https://github. com/webstorms/ NeuralPred | GitHub, NeuralPred |
| Singer Y, Taylor L, Willmore BDB, King AJ, Harper NS | 2023 | Hierarchical temporal prediction captures motion processing along the visual pathway. Figshare ID natural.zip | https://figshare.com/ articles/dataset/ Natural_movies/ 24265498 | figshare, 24265498 |

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
