## [Editor Report]

This valuable work shows similarities between a multilayer, convolutional neural network trained to predict its next input and physiological features of visual processing in the brain. These solid results build on the authors' previous work and compare the match to real visual processing obtained by a hierarchical predictive network to that obtained by several other popular artificial neural networks. This work will be of interest to systems neuroscientists as well as computer scientists looking to make connections between normative theories of neural organization and training objectives in machine learning.

---

## [Decision Letter]

**Decision letter after peer review:**

Thank you for submitting your article "Hierarchical temporal prediction captures motion processing along the visual pathway" for consideration by *eLife*. Your article has been reviewed by 2 peer reviewers, and the evaluation has been overseen by a Reviewing Editor and Joshua Gold as the Senior Editor. The following individual involved in review of your submission has agreed to reveal their identity: Dan Yamins (Reviewer #1).

The reviewers have discussed the reviews with one another and the Reviewing Editor has drafted this decision to help you prepare a revised submission.

Summary:

This paper expands on a predictive-coding-like unsupervised learning procedure and applies it to natural videos, training a multilayer, convolutional neural network with the objective of predicting the next input. The resulting network is qualitatively compared to several known neurophysiological features of the (dorsal) visual pathway, focusing on V1, but also comparing model results to receptive field properties of the LGN and area MT. The authors find center surround, simple, and complex cell behavior like that found in the retina, LGN, and V1. They also find units sensitive to drifting plaids. Matches between the predictive performance of this network are better than for a basic autoencoder and also better than several other models that ablate aspects of the proposed encoding rule. The paper hypothesizes that prediction might be a fundamental principle of self-organization in the visual system.

Essential revisions:

This paper advances a reasonable hypothesis about a very important goal: developing an unsupervised neural network model of the visual system. However, the current draft is not ready for publication because empirical validation of the model and comparison to equally reasonable alternatives are weak. The following revisions were deemed essential by the reviewing team:

1) Key controls are missing: a comparison between the results of the present work and other obvious alternative hypotheses should be included. Here are five comparisons that should be made, in order of increasing power:

a) A sparse autoencoder. A stacked autoencoder is tested, but it's not said whether this model was regularized with an activity sparsity penalty on the bottleneck or not. If not, this should be done.

b) A model based on temporal continuity / slowness rather than predictability. It's likely that pretty much the same spatial tuning properties would emerge, but that subtle differences in the temporal dynamics would be detectable. This would be particularly interesting in the light of a recent paper by Weghenkel and Wiskott [1] (2018, Neural Computation), which has shown no advantage of predictability over slowness on a large set of real world data sets.

c) A Gabor-wavelet-based model of V1. A good example of that is the Berkeley Wavelet Transform (BTW) in Willmore et al. [2]. Obviously this might not have the temporal processing facets of the authors' model, but the main comparisons made to data in Figure 6 don't require this. A very basic question is: how much better (or worse?) is the authors' model as a model of V1, as compared to the hard-coded (non-learned) "standard" model of V1 that the BWT represents?

d) The PredNet [3] and PredRNN [4] and PredRNN++ [5] models. These are predictive coding models similar to the one the authors propose. The authors of course do cite [3], but do not compare to its results. They definitely should. Merely noting that Prednet "has not been demonstrated [to] capture the phenomena that we describe …. " does not mean that PredNet *wouldn't* capture these phenomena. Being fairly familiar with PredNet , we imagine that it could indeed capture these features. Showing that it does not, and that the current proposal is thus a better fit to the data, is a burden that is clearly on the authors in the case. The code for prednet is publicly released and could easily be downloaded and run by the authors (trained on their training set). We also suggest strongly that the authors look at PredRNN as well, which may be substantially better than PredNet.

e) The early layers of a supervised deep neural network. This is an important baseline -- how much better (if at all) are these authors' models at matching V1 data than the supervised network, which is (as the authors correctly point out) obviously trained in a deeply unbiologial way? Work such as that of Cadena et al. [6] clearly shows that the supervised network actually gives very reasonable results in V1 -- in fact, it's the state-of-the-art model of V1 in the literature, at least to our knowledge. How much better is their unsupervised network than this model at matching V1? If it's not better, then how big is the gap, and why would it exist? (If a supposedly much more biologically-correct model isn't substantially better than the unbiological one at matching biology data, has it really contributed a major advance?)

One thing that will naturally come up in making a comparison like this is that the authors have chosen a particular imageset (moving animals) for training. Perhaps it will arise that their model is not trained in a sufficiently general way to compare favorably with a model trained on a large dataset like ImageNet. (Of course, this is still a fair comparison since neither their model nor the deepnet model would have been trained on the set used for testing, e.g. oriented gratings, or some other set of natural or naturalistic stimuli.) To claim that a major advance in modeling V1 has been made, something along these lines needs to be included. Perhaps the authors could use large video datasets like Kinetics [7] or Moments in Time [8] or the Chicago Motion Database (https://cmd.rcc.uchicago.edu) as a replacement for their animals dataset.

2) The comparisons to neural data are weak and qualitative. Improving that w.r.t. V1 *or* MT would be a major and sufficient revision for publication. The network learns mostly center surround, simple and complex cell behavior, which is conventionally assigned to retina/LGN/V1, but other features of V1 and higher-order visual neurons are not observed. No end- or side-inhibition has been observed, and no object-sensitive units. Only a very small fraction of units sensitive to drifting plaids were found in the 4th stack/layer. Given that, billing the model as a candidate for explaining the whole visual system seems much overstated. The authors could capitalize more on the dynamics of the receptive fields in V1 or MT, which was an interesting result not so often obtained by other models, but that has been investigated much less and was not compared to experimental data (apart from motion sensitivity). In whichever way possible, more detailed comparisons to V1 or MT data are needed. Our suggestions are:

a-i) For V1, the comparisons in Figure 6 are fine, and represent a good first step at comparing models to the data in a gross way. However, it would be great if a slightly more quantitative approach was taken -- e.g. measuring model-similarity in some quantified way, especially to compare between the author's preferred model and the controls suggested in 1) above.

a-ii) Also for V1: A much stronger comparison would be to do something as in Cadena et al. [6]. Specifically, Cadena et al. build a neuron-by-neuron regression model from their model to real V1 neurons, on a large set of real-world and naturalistic images. That work shows that, on this type of high-resolution comparison, there is something substantially better as a model of V1 than the standard hand-coded gabor BWT -- namely, the early intermediate layer of a categorization-trained deepnet. The state of the field has now moved to a point where models are being separated not by coarse measures like what is shown in Figure 6, but rather these much more detailed, real-world-stimulus-based metrics. We think the authors need to address comparisons at this level of resolution, or else it's really hard to know whether their model has made any substantive advance. It's not clear whether the data from Cadena et al. is readily available (though we suspect that Andreas Tolias, the data generator for that paper, would provide it for this purpose if asked). However, there is (or at least used to be) publicly available data from the Neural Prediction Challenge -- definitely worth getting this or similar data from Jack Gallant at Berkeley. (Or any other source that would allow for a much more direct model-to-neuron prediction assessment across naturalistic stimuli.)

b) For MT: the comparison here is quite thin. What the authors have done seems to barely support the claim that their hierarchical model "can capture how tuning properties change across multiple levels of the visual system". More needs to be done here. Several papers have shown such things, mostly (as the authors note) based on supervised models. E.g [9-11] show comparisons of various intermediate layers of a NN to V1, V4, PIT, AIT areas in the ventral visual pathway. An unsupervised model that did the equivalent of this would be a significant advance. To make a claim like what the authors are saying in this draft, there needs to be some equally strong data comparison, but with MT data. Shinji Nishimoto and Jack Gallant have collected data that would be useful for this comparison, but it's not clear whether it would be easy to get access to that or similar MT data.

Comparing to coding in the ventral stream might be an alternative if MT data are not available. V4 and IT data is easily available from Jim DiCarlo (e.g. the data for [9-10]). The authors could definitely check out how well their higher model layers regressed those data and see if they could sustain a claim about matching "multiple levels of the visual system". (But perhaps it would be an unfair comparison? Do the authors think their model would have any power for describing the ventral pathway?)

If these more detailed comparisons cannot be made, the claims about matching "multiple levels of the visual system" must be removed or significantly modified.

[1] Weghenkel, Björn, and Laurenz Wiskott. "Slowness as a proxy for temporal predictability: An empirical comparison." Neural computation 30, no. 5 (2018): 1151-1179.

[2] Willmore, Ben, Ryan J. Prenger, Michael C-K. Wu, and Jack L. Gallant. "The berkeley wavelet transform: a biologically inspired orthogonal wavelet transform." Neural computation 20, no. 6 (2008): 1537-1564.

[3] Lotter, William, Gabriel Kreiman, and David Cox. "Deep predictive coding networks for video prediction and unsupervised learning." arXiv preprint arXiv:1605.08104 (2016).

[4] Wang, Yunbo, Mingsheng Long, Jianmin Wang, Zhifeng Gao, and S. Yu Philip. "Predrnn: Recurrent neural networks for predictive learning using spatiotemporal lstms." In Advances in Neural Information Processing Systems, pp. 879-888. 2017.

[5] Wang, Yunbo, Zhifeng Gao, Mingsheng Long, Jianmin Wang, and Philip S. Yu. "Predrnn++: Towards a resolution of the deep-in-time dilemma in spatiotemporal predictive learning." arXiv preprint arXiv:1804.06300 (2018).

[6] Cadena, Santiago A., George H. Denfield, Edgar Y. Walker, Leon A. Gatys, Andreas S. Tolias, Matthias Bethge, and Alexander S. Ecker. "Deep convolutional models improve predictions of macaque V1 responses to natural images." PLoS computational biology 15, no. 4 (2019): e1006897.

[7] https://deepmind.com/research/open-source/kinetics

[8] http://moments.csail.mit.edu/

[9] Yamins, Daniel LK, Ha Hong, Charles F. Cadieu, Ethan A. Solomon, Darren Seibert, and James J. DiCarlo. "Performance-optimized hierarchical models predict neural responses in higher visual cortex." Proceedings of the National Academy of Sciences 111, no. 23 (2014): 8619-8624.

[10] Nayebi, Aran, Daniel Bear, Jonas Kubilius, Kohitij Kar, Surya Ganguli, David Sussillo, James J. DiCarlo, and Daniel L. Yamins. "Task-Driven convolutional recurrent models of the visual system." In Advances in Neural Information Processing Systems, pp. 5290-5301. 2018.

[11] Khaligh-Razavi, Seyed-Mahdi, and Nikolaus Kriegeskorte. "Deep supervised, but not unsupervised, models may explain IT cortical representation." PLoS computational biology 10, no. 11 (2014): e1003915.

---

## [Author Response]

Essential revisions:This paper advances a reasonable hypothesis about a very important goal: developing an unsupervised neural network model of the visual system. However, the current draft is not ready for publication because empirical validation of the model and comparison to equally reasonable alternatives are weak. The following revisions were deemed essential by the reviewing team:1) Key controls are missing: a comparison between the results of the present work and other obvious alternative hypotheses should be included. Here are five comparisons that should be made, in order of increasing power:

The original purpose of this research advance was to demonstrate that the unsupervised temporal prediction model described in our original *eLife* paper (Singer et al., 2018) as an explanation for the receptive field properties of neurons in primary visual (and auditory) cortex, could be extended to a hierarchical form that reproduced receptive field properties at different levels of the visual pathway. This was not intended to be a comprehensive quantitative comparison of different models of the visual pathway, which we think would go beyond the requirements of a research advance.

However, we certainly see the merits of the reviewers’ suggestions, and we have now made many of the requested changes. This has been a very substantial undertaking because it required the development of two new hierarchical models based on different principles (sparsity and slowness), adaptation of previously published models, and the building of the model testing framework.

As set out in the following, our new results demonstrate that our temporal prediction model is a leading contender among other principles for explaining the organization of the visual pathway. Attempting to identify a single winner would be misleading, because of variation between the models in their size, aims and scope, as well as implementational factors such as variation in performance over different training sets and the possibility that hyperparameters may not be perfectly optimized. Accounting fully for all of these factors is well beyond the scope of this study.

With this in mind, we have implemented the majority of the models that were requested and compared them quantitatively to the hierarchical temporal prediction model. We have focused on what we view as the most important and challenging comparison, the prediction of neural responses to natural stimuli.

As the additions and edits that we have made to the Results, Discussion and Methods are extensive and relevant to both Essential Revisions 1 and 2a, we will first respond to each point in 1 and 2a, and then post the edits in one large block after the response to point 2a.

a) A sparse autoencoder. A stacked autoencoder is tested, but it's not said whether this model was regularized with an activity sparsity penalty on the bottleneck or not. If not, this should be done.

We have now implemented a stacked autoencoder with activity sparsity. To facilitate comparison, we have given this the same structure as our temporal prediction model, except that we changed the objective function of each stack to that of an autoencoder with sparse activity.

b) A model based on temporal continuity / slowness rather than predictability. It's likely that pretty much the same spatial tuning properties would emerge, but that subtle differences in the temporal dynamics would be detectable. This would be particularly interesting in the light of a recent paper by Weghenkel and Wiskott [1] (2018, Neural Computation), which has shown no advantage of predictability over slowness on a large set of real world data sets.

We have implemented a version of this principle as a stacked network, whose activity is constrained to vary slowly over time and to be decorrelated over units. To facilitate comparison, we have given this the same structure as our temporal prediction model, except that we changed the objective function of each stack to that of finding slow features.

As an aside, there are various differences between the predictability models used in Weghenhal and Wiskott (2018) and our temporal prediction model. Notably, the models in Weghenkel and Wiskott find those features whose activation is predictable over time (predictable features), whereas temporal prediction finds those features that are predictive of future input (predictive features). While related, these are not the same. For our edits to the text on this point, please see the Discussion.

c) A Gabor-wavelet-based model of V1. A good example of that is the Berkeley Wavelet Transform (BTW) in Willmore et al. [2]. Obviously this might not have the temporal processing facets of the authors' model, but the main comparisons made to data in Figure 6 don't require this. A very basic question is: how much better (or worse?) is the authors' model as a model of V1, as compared to the hard-coded (non-learned) "standard" model of V1 that the BWT represents?

We adapted the Berkeley Wavelet Transform for this comparison.

d) The PredNet [3] and PredRNN [4] and PredRNN++ [5] models. These are predictive coding models similar to the one the authors propose. The authors of course do cite [3], but do not compare to its results. They definitely should. Merely noting that Prednet "has not been demonstrated [to] capture the phenomena that we describe …. " does not mean that PredNet *wouldn't* capture these phenomena. Being fairly familiar with PredNet , we imagine that it could indeed capture these features. Showing that it does not, and that the current proposal is thus a better fit to the data, is a burden that is clearly on the authors in the case. The code for prednet is publicly released and could easily be downloaded and run by the authors (trained on their training set). We also suggest strongly that the authors look at PredRNN as well, which may be substantially better than PredNet.

We have adapted PredNet for this comparison. One distinct advantage that PredNet (and PredRNN) have over all the other models (including the temporal prediction model) is that they are recurrent, an important property shared with the brain. Arguably, this is therefore not a fair comparison. We are currently developing a recurrent version of our hierarchical temporal prediction model, and we certainly plan to make quantitative comparisons with other recurrent models. For now, we have included PredNet as a model for comparison with our existing model, but not PredRNN or PredRNN++, as we feel that a future publication would be more appropriate for a fair comparison between recurrent models.

e) The early layers of a supervised deep neural network. This is an important baseline -- how much better (if at all) are these authors' models at matching V1 data than the supervised network, which is (as the authors correctly point out) obviously trained in a deeply unbiologial way? Work such as that of Cadena et al. [6] clearly shows that the supervised network actually gives very reasonable results in V1 -- in fact, it's the state-of-the-art model of V1 in the literature, at least to our knowledge. How much better is their unsupervised network than this model at matching V1? If it's not better, then how big is the gap, and why would it exist? (If a supposedly much more biologically-correct model isn't substantially better than the unbiological one at matching biology data, has it really contributed a major advance?)

We have adapted the visual geometry group (VGG) model for this comparison. Please note, however, that this is a supervised model, which was fitted to human behavioral data (labeled images), rather than a normative model in the strict sense (where the model is trained according to a statistical principle, without fitting to behavioral or neural data). Having these behavioral data provides quantitative measurements of the outputs of neural systems that can be used to constrain the model, which is not available to the other models that are being compared here, such as temporal prediction.

One thing that will naturally come up in making a comparison like this is that the authors have chosen a particular imageset (moving animals) for training. Perhaps it will arise that their model is not trained in a sufficiently general way to compare favorably with a model trained on a large dataset like ImageNet. (Of course, this is still a fair comparison since neither their model nor the deepnet model would have been trained on the set used for testing, e.g. oriented gratings, or some other set of natural or naturalistic stimuli.) To claim that a major advance in modeling V1 has been made, something along these lines needs to be included. Perhaps the authors could use large video datasets like Kinetics [7] or Moments in Time [8] or the Chicago Motion Database (https://cmd.rcc.uchicago.edu) as a replacement for their animals dataset.

Our dataset consisted of videos of different animals in motion in diverse natural environments, which is a relatively rich dataset and close to the kind of visual stimuli that would often occur in the wild. However, we agree with the reviewers that the diversity of the dataset could be increased still further, most notably with other kinds of moving forms and with forward motion. To this end, for our new figure (Figure 9), we have now trained the temporal prediction model on a new dataset of even more diverse stimuli. We recorded this new dataset ourselves and it includes not just animals in motion, but other forms of motion such as rolling balls or trees moving in the wind, as well as diverse camera dynamics, such as forward motion for visual flow, panning, and stillness.

2) The comparisons to neural data are weak and qualitative. Improving that w.r.t. V1 *or* MT would be a major and sufficient revision for publication. The network learns mostly center surround, simple and complex cell behavior, which is conventionally assigned to retina/LGN/V1, but other features of V1 and higher-order visual neurons are not observed. No end- or side-inhibition has been observed, and no object-sensitive units. Only a very small fraction of units sensitive to drifting plaids were found in the 4th stack/layer. Given that, billing the model as a candidate for explaining the whole visual system seems much overstated. The authors could capitalize more on the dynamics of the receptive fields in V1 or MT, which was an interesting result not so often obtained by other models, but that has been investigated much less and was not compared to experimental data (apart from motion sensitivity). In whichever way possible, more detailed comparisons to V1 or MT data are needed. Our suggestions are:a-i) For V1, the comparisons in Figure 6 are fine, and represent a good first step at comparing models to the data in a gross way. However, it would be great if a slightly more quantitative approach was taken -- e.g. measuring model-similarity in some quantified way, especially to compare between the author's preferred model and the controls suggested in 1) above.a-ii) Also for V1: A much stronger comparison would be to do something as in Cadena et al. [6]. Specifically, Cadena et al. build a neuron-by-neuron regression model from their model to real V1 neurons, on a large set of real-world and naturalistic images. That work shows that, on this type of high-resolution comparison, there is something substantially better as a model of V1 than the standard hand-coded gabor BWT -- namely, the early intermediate layer of a categorization-trained deepnet. The state of the field has now moved to a point where models are being separated not by coarse measures like what is shown in Figure 6, but rather these much more detailed, real-world-stimulus-based metrics. We think the authors need to address comparisons at this level of resolution, or else it's really hard to know whether their model has made any substantive advance. It's not clear whether the data from Cadena et al. is readily available (though we suspect that Andreas Tolias, the data generator for that paper, would provide it for this purpose if asked). However, there is (or at least used to be) publicly available data from the Neural Prediction Challenge -- definitely worth getting this or similar data from Jack Gallant at Berkeley. (Or any other source that would allow for a much more direct model-to-neuron prediction assessment across naturalistic stimuli.)

We agree and have opted to focus on the stronger comparison (as set out in reviewer comment a-ii). We have added the following sections to the manuscript, which address the points made above in response to points 1 and 2a.

We have added the following new section and Figure to the Results (pages 22-24):

“Predicting neural responses to natural stimuli

A standard method to assess a model’s capacity to explain neural responses is to measure how well it can predict neural responses to natural stimuli. We did this for our model for two datasets. The first comprised single-unit neural responses recorded from awake macaque V1 to images of natural scenes (Cadena et al., 2019). The second, particularly relevant to our model, consisted of multiunit responses from anesthetized macaque V1 to movies of natural scenes (Nahaus and Ringach, 2007; Ringach and Nahaus, 2009). Estimating neural responses to such dynamic natural stimuli has received less attention in models of visual processing.

[…]

On predicting the neural responses to images (Figure 9A), relative to the temporal prediction model, the BWT and autoencoder models performed slightly worse (bootstrap, n = 166, p < 10^-4^ and p < 10^-4^, respectively, see Methods) and the VGG, PredNet and slowness models performed slightly better (bootstrap, n = 166, p < 10^-4^, p < 10^-4^ and p = 5.3×10^-3^, respectively). However, for the dynamic visual stimuli, the results were quite different (Figure 9B). The BWT, PredNet, autoencoder, and slowness models all performed worse than the temporal prediction model (bootstrap, n = 23, p = 1.6×10^-2^, p = 2×10^-4^, p = 2.5×10^-3^ and p < 10^-4^, respectively), with only the VGG model being not significantly different (bootstrap, n = 23, n.s.). Hence, the temporal prediction model, an unsupervised model trained on unlabeled data with no explicit constraints imposed by either neural or behavioral data, rivals a supervised model (VGG) trained using a labeled behavioral dataset in predicting neural responses in V1 to dynamic stimuli. Thus, the temporal prediction model is a reasonable contender as an unsupervised model of V1 responses to natural stimuli, particularly dynamic natural stimuli.”

We have also added the following section to the Discussion (pages to 33-34):

“Quantitative comparison to different models

Because the visual system has evolved to process natural stimuli, an important consideration in whether a computational model captures visual processing is whether it can predict neural responses to such stimuli. To this end, we estimated how well the hierarchical temporal prediction model can predict responses in awake macaque V1 to movies and images of natural scenes. Of particular interest are responses to movies, as our model is focused on the neural processing of dynamic stimuli. We compared the temporal prediction model to other models of visual system organization. Two of these models have identical structure to the temporal prediction model, but implement different underlying principles; one model estimates the current rather than future input alongside a sparse activity constraint (sparse coding principle) and the other finds slowly-varying features (slowness principle). We also made comparisons with three other key models of visual processing, the PredNet model, which is based on the principle of predictive coding, a wavelet model (BWT), inspired by the classic Gabor model, and a deep supervised model trained on image recognition (VGG).

[…]

We found that the performance of the hierarchical temporal prediction model at predicting the responses of V1 neurons to images was not significantly different from leading models such as the VGG model. Importantly, for the V1 responses to movies, the temporal prediction model outperformed all the other models, except for the VGG model, which it performed as well as. However, the VGG model has the advantage of labeled data, and was also trained on far more data than our model, whereas the temporal prediction model is unsupervised. Our findings therefore indicate that temporal prediction remains a leading contender as an underlying normative unsupervised principle governing the neural processing of natural visual scenes.”

Furthermore, we have added the following section to the Methods (pages 58-67):

“Quantitative model comparison for neural response prediction

We compared how well the temporal prediction model and five other models could predict neural responses. The models that we compared to the temporal prediction model were the following: the visual geometry group (VGG) model – a spatial deep convolutional model trained in a supervised manner on labeled data for image recognition, which has been shown to predict well neural responses in V1 (Cadena et al., 2019); the Berkeley wavelet transform model – a descriptive spatial model of V1, and three unsupervised spatiotemporal models – a leading predictive coding model (PredNet), a slowness model and an autoencoder model with sparse activity. The models were compared for their capacity to predict neural responses to images and movies of natural scenes.

[…]

Statistical tests

Statistical comparison of the different models was performed using a bootstrapping method (Zhuang et al., 2021). To calculate the probability *p* of model A performing better/worse statistically than model B, we sampled prediction scores over the neurons (paired, with replacement) for each model and calculated their respective means. We did this 10,000 times, counting the number of times model A had a larger mean than model B. Finally, this count was divided by 10,000 to obtain the probability *p* of model A having a larger mean than model B, with 1-p being the probability of model B having a larger mean than model A.”

b) For MT: the comparison here is quite thin. What the authors have done seems to barely support the claim that their hierarchical model "can capture how tuning properties change across multiple levels of the visual system". More needs to be done here. Several papers have shown such things, mostly (as the authors note) based on supervised models. E.g [9-11] show comparisons of various intermediate layers of a NN to V1, V4, PIT, AIT areas in the ventral visual pathway. An unsupervised model that did the equivalent of this would be a significant advance. To make a claim like what the authors are saying in this draft, there needs to be some equally strong data comparison, but with MT data. Shinji Nishimoto and Jack Gallant have collected data that would be useful for this comparison, but it's not clear whether it would be easy to get access to that or similar MT data.Comparing to coding in the ventral stream might be an alternative if MT data are not available. V4 and IT data is easily available from Jim DiCarlo (e.g. the data for [9-10]). The authors could definitely check out how well their higher model layers regressed those data and see if they could sustain a claim about matching "multiple levels of the visual system". (But perhaps it would be an unfair comparison? Do the authors think their model would have any power for describing the ventral pathway?)If these more detailed comparisons cannot be made, the claims about matching "multiple levels of the visual system" must be removed or significantly modified.

We agree that the number of pattern-motion selective neurons in our model is small, but we still think it is interesting that they are observed. We are now developing a recurrent version of our hierarchical temporal prediction model for which early results are showing a much greater number of pattern-motion selective neurons. This represents a major extension of our existing model and is therefore more appropriate for a future publication. Given that the current paper is being considered as a “research advance” that builds on the findings of our original 2018 *eLife* paper, we have opted to focus more on the progression from retina/LGN to the granular layer of V1 to the superficial layers of V1, and weakened our previous statements about higher areas. Our paper is now more focused on how the model reproduces properties of the retina and granular and supragranular V1. We have made the following changes throughout the manuscript.

In the Abstract we changed “multiple” to the substantially more circumspect “at least two”:

“Here we show that hierarchical application of temporal prediction can capture how tuning properties change across at least two levels of the visual system.”

In the Introduction we changed “multiple” to “several” and “visual motion pathway” to “visual pathway” (line 74):

“The capacity of this model to explain spatiotemporal tuning properties at several levels of the visual pathway using iterated application of a single process, suggests that optimization for temporal prediction may be a fundamental principle of sensory neural processing.”

In the Results we added the following sentences with regard to Figure 7 (pages 16-17):

“A clear progression from non-orientation-selective, to simple-cell like, to complex-cell like is seen in the units as one progresses up the stacks of the model (Figure 7). This bears some resemblance to the progression from the retina and LGN, which has few orientation-selective neurons, at least in cats and monkeys (Barlow, 1953; Kuffler, 1953; Shapley and Perry, 1986), to the geniculorecipient granular layer of V1, which tend to show more simple cells, to the superficial layers of V1 where more complex cells have been found (Ringach 2002, Cloherty and Ibbotson 2015). This is also consistent with the substantial increase in the proportion of complex cells from V1 to extrastriate visual areas, such as V2 (Cloherty and Ibbotson 2015).”

We also substantially weakened claims in the Conclusions (page 34-35):

“Previous studies using principles related to temporal prediction have produced non-hierarchical models of retina or primary cortex alone and demonstrated retina-like RFs (Ocko et al., 2018), simple-cell like RFs (Chalk et al., 2018; Singer et al., 2018), or RFs resembling those found in primary auditory cortex (Singer et al., 2018). However, any general principle of neural representation should be extendable to a hierarchical form and not tailored to the region it is attempting to explain. Here we show that temporal prediction can indeed be made hierarchical and so reproduce the major motion-tuning properties that emerge along the visual pathway from retina and LGN to V1 granular layers to V1 superficial layers, and potentially some properties of the extrastriate dorsal cortical pathway. This model captures not only linear tuning features, such as those seen in center-surround retinal neurons and direction-selective simple cells, but also nonlinear features seen in complex cells, with some indication that it may also reproduce properties of the pattern-sensitive neurons that are found in area MT. The capacity of our hierarchical temporal prediction model to account for many tuning features at several levels of the visual system suggests that the same framework may well explain many more features of the brain than we have so far investigated. Iterated application of temporal prediction, as performed by our model, could be used to make predictions about the tuning properties of neurons in brain pathways that are much less well understood than those of the visual system. Our results suggest that, by learning behaviorally-useful features from dynamic unlabeled data, temporal prediction may represent a fundamental coding principle in the brain.”